# Analysis of Phlebotomine sandflies in Laos from 2014–2024: Inventory, description of a new species, screening for *Leishmania* and detection of *Trypanosoma*

Khamsing Vongphayloth[1,2]*, Tamalee Roberts[3,4], Jodi M. Fiorenzano[5], Noel Cote[5], Irina V. Etobayeva[5], Aphaphone Adsamouth[3], Matthew T. Robinson[3,4], Khaithong Lakeomany[1], Veaky Vungkyly[1], Phonesavanh Luangamath[1], Somsanith Chonephetsarath[1], Fano José Randrianambinintsoa[2], Andrew G. Letizia[5], Philippe Buchy[1], Paul T. Brey[1], Jérôme Depaquit[2,6]

**1** Institut Pasteur du Laos, Ministry of Health, Laboratory of Medical Entomology and Vector-Borne Diseases, Sisattanak District, Vientiane, Lao PDR, **2** Université de Reims Champagne-Ardenne, UR ESCAPE-USC ANSES PETARD, Reims, France, **3** Lao-Oxford-Mahosot Hospital-Wellcome Trust Research Unit (LOMWRU), Mahosot Hospital, Vientiane, Lao People's Democratic Republic, **4** Centre for Tropical Medicine and Global Health, Nuffield Department of Medicine, University of Oxford, United Kingdom, **5** Science Directorate, U.S. Naval Medical Research Unit INDO PACIFIC, Sembawang, Singapore, **6** Centre Hospitalo-Universitaire de Reims, Pôle de Biologie Territoriale, Laboratoire de Parasitologie-Mycologie, Reims, France

* k.vongphayloth@pasteur.la

## Abstract

### Introduction

Phlebotomine sandflies are the principal vectors of leishmaniasis. Laos is a land-locked country bordering Thailand, where autochthonous cases of leishmaniasis have been reported. However, the status of leishmaniases in Laos is unknown due to a lack of testing. In the past decades, very few studies on sandflies and sandfly-borne pathogens have been conducted in Laos. Therefore, the knowledge of sandfly diversity, distribution, and their related pathogens is lacking. We hypothesized that both known and putative sandfly-borne pathogens may silently circulate in Laos but remain undetected. Herein, we aimed to report species diversity data and *Leishmania* detection among sandflies collected from seven provinces of Laos.

### Methods

Sandflies were collected using CDC light traps from various habitats, including caves and peri-caves in karstic areas, domestic animal sheds, forests, and plantations, across seven provinces in northern Laos. Sandfly collections were conducted between 2014 and 2024 from different seasons. Sandflies were identified by morphological and molecular techniques. PCR targeting the ITS1 region was used to screen

**Data availability statement:** Sequences of Sandfly Cyt-b obtained from this study were deposited in GenBank under accession numbers PV054613– PV054752). Sequences of Trypanosoma spp. detected from this study were deposited in the GenBank database (PV034524 - PV034547). Type specimens. The holotype female (voucher SF22-EX651) of Sergentomyia sutherlandi n. sp. and one female paratype (voucher SF22-EX695) have been deposited at the Terrestrial Arthropods Collection of the Muséum national d'Histoire naturelle (MNHN, Paris) under the inventory numbers MNHN-ED-11547 and MNHN-ED-11548.

**Funding:** This work was partial supported by the Global Emerging Infections Surveillance program (GEIS) work units (PROMIS ID# P0089_22_N2 to JMF, P0071_23_N2 to NC, and P0126_24_N2 to NC, IVE) as a component of the Armed Forces Health Surveillance Branch (AFHSB) awarded to the U.S Naval Medical Research Unit INDO PACIFIC (NAMRU INDO PACIFIC), and by the French Ministry of Higher Education, Research and Innovation via the Institut Pasteur du Laos (MESRI.K-23-24 to KV). Leishmania testing was funded under ESCMID Research Grant 2021 (awarded to TR). TR and MTR are funded by the Wellcome Trust [220211/Z/20/Z]. The funders had no role in study design, data collection and analysis, decision to publish, or preparation of the manuscript. For the purpose of Open Access, the author has applied a CC BY public copyright license to any Author Accepted Manuscript version arising from this submission.

**Competing interests:** The authors have declared that no competing interests exist.

for *Leishmania* in the sandfly samples. Though a specific *Trypanosoma* PCR was not used, this PCR is also able to detect the ITS1 region in some *Trypanosoma* spp.

## Results

A total of 3,857 sandflies from 25 species belonging to five genera were collected and examined: *Chinius eunicegalatiae*, *Idiophlebotomus longiforceps*, *Phlebotomus barguesae*, *Ph. breyi*, *Ph.* (*Adlerius*) sp., *Ph. kiangsuensis*, *Ph. mascomai*, *Ph. seowpohi*, *Ph. shadenae*, *Ph. sinxayarami*, *Ph. stantoni*, *Sergentomyia anodontis*, *Se. bailyi*, *Se. barraudi* group, *Se. brevicaulis*, *Se. dvoraki*, *Se. hivernus*, *Se. khawi*, *Se. perturbans*, *Se. phasukae*, *Se. sutherlandi* n. sp., *Se. sylvatica*, *Se. tambori*, *Se. gemmea*-like (*Se.* sp 1) and *Grassomyia indica*. The highest diversity of sandflies was found in karstic areas where sandflies were collected from cave and peri-cave areas. One new sandfly species, *Se. sutherlandi* n. sp. is described. We also report for the first time in Southeast Asia a *Phlebotomus* female belonging to the subgenus *Adlerius* and we also discuss about the taxonomy of *Sergentomyia brevicaulis*. Although no *Leishmania* DNA was detected from screened sandflies, unknown *Trypanosoma* species were detected from 24 individual sandflies: *Chinius eunicegalatiae* (n = 22) and *Idiophlebotomus longiforceps* (n = 2) using a PCR assay that was primarily meant to screen for *Leishmania*.

## Conclusion

This study reveals a notable diversity of sandfly species across seven provinces in Laos, with the highest species richness observed in karstic cave environments. While no *Leishmania* DNA was detected, the unexpected identification of unknown *Trypanosoma* species in *Chinius eunicegalatiae* and *Idiophlebotomus longiforceps* suggests the potential presence of unrecognized trypanosomatid in the region. These findings underscore the need for broader geographic surveillance and increased sampling efforts to improve our understanding of sandfly ecology and the pathogens they may carry in Laos.

### Author summary

Phlebotomine sandflies are best known as insect vectors of arboviruses and leishmaniasis, a parasitic disease of global health concern. While neighboring Thailand has reported locally acquired cases, the status of leishmaniasis in Laos has remained uncertain due to limited surveillance. In this study, sandflies from seven provinces in Laos were surveyed and tested for the presence of *Leishmania* parasites. The survey revealed 25 sandfly species, with the greatest diversity found in karstic cave environments, and led to the discovery of a new species, *Sergentomyia sutherlandi*. Although *Leishmania* parasites were not detected, DNA from an unknown *Trypanosoma* species was unexpectedly found in two

sandfly species using a molecular test originally designed to detect *Leishmania*. These findings highlight both the rich, previously undocumented sandfly biodiversity in Laos and the potential presence of other parasitic pathogens. This work expands the foundation for future research on vector-borne diseases in Southeast Asia and underscores the need for broader entomological and pathogen surveillance in underexplored regions.

## Introduction

The emergence of new and the resurgence of old known vector-borne pathogens can be associated with several factors, including adaptation and change of microorganisms, habitat changes, globalization, tourism, and travel [1,2]. Phlebotomine sandflies (Diptera: Psychodidae) include a diverse group of vectors that vary widely in geographic distribution, ecology, and pathogens they transmit. Most of the medically important species mainly belong to the genera *Lutzomyia, Psychodopygus, Nyssomyia,* and *Pintomyia* in the New World and *Phlebotomus* in the Old World [3,4]. Although sandflies readily bite humans, they are primarily opportunistic feeders [5], taking a blood meal from almost any cold-blooded animal or warm-blooded mammal or bird they encounter.

Southeast Asia is known as a global hotspot of biodiversity [6]. But very little is known about the diversity of sandflies and their related pathogens or diseases that they transmit in this region. Before 1996, only imported cases of leishmaniasis had been reported from this region, predominantly within migrant populations. However, the first autochthonous human case of leishmaniasis was reported in Thailand in 1996, and since then, over 100 cases have been reported with the predominant species *Leishmania* (*Mundinia*) *martiniquensis* and *L. (Mundinia) orientalis* [7–10]. Within Southeast Asian countries, sandfly taxonomy has been most widely studied in Thailand. A total of 37 sandfly species have been reported in Thailand with widespread distribution across the country [11–14]. *Sergentomyia barraudi* and *Se. iyengari* groups (possibly *Se. khawi* but misidentified as *Se. gemmea*) [15], have been found to harbor *Leishmania* DNA (identified by PCR). Due to the biodiversity and unique ecological conditions, such as its varied habitats and climate, Lao PDR (Laos) is likely to have a diverse and unique population of sandflies and potential pathogens that they vector.

Laos, a landlocked country in Southeast Asia, has become land-linked through a high-speed railway connecting Vientiane Capital to Kunming, China. However, diagnostic tools and surveillance systems for infectious diseases are lacking. There have been limited studies on the prevalence of sandfly-borne diseases, especially leishmaniasis, in Laos [16]. Moreover, the data on the diversity of sandfly fauna of Laos is very limited. Sandfly studies in Laos were carried out predominantly before the 1970s by Raynal (1936), Parrot and Clastrier (1952), and Quate (1962) [17–19]. Since then, there has only been one report of a new species of sandfly in Laos, which was in 2010 [20]. From these works, only nine sandfly species have been identified in Laos. To fill the gap in knowledge on sandfly species, their distribution and related sandfly-borne pathogens in Laos, the Institut Pasteur du Laos (IPL) has performed sandfly and sandfly-related pathogen surveillance studies from 2012 to 2023. These investigations identified two new sandfly species within a new subgenus in 2023 [21], and later two more were added in 2024 from sandflies collected in 2019 from Vientiane province [22].

Herein we report the inventory of sandfly species diversity and results of primarily screening for *Leishmania* from sandfly samples collected from seven provinces in Laos between 2014–2024.

## Methods

### Sandfly collection sites

Sandflies previously collected and stored at the IPL were examined and included in this study. These samples were collected between 2014 and 2024 from seven provinces of northern Laos including Bokeo, Luangnamtha, Luangphabang, Vientiane Capital, Vientiane province, Xayaboury, and Xiengkhouang (Fig 1). The map of the sampling sites was created using QGIS software and open-source data. The Lao administrative shapefile data was downloaded from the

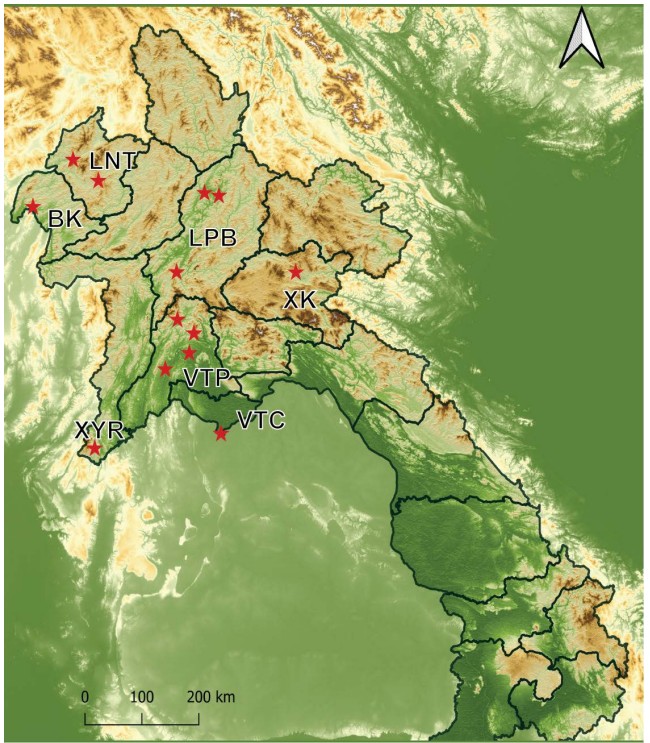

**Fig 1. Collection sites across Laos.** Red stars represent the collection sites in this study. BK: Bokeo, LNT: Luangnamtha, LPB: Luangphabang, VTP: Vientiane province, VTC: Vientiane Capital, XYR: Xayaboury, and XK: Xiengkhouang. The map of the sampling sites was created using QGIS software and open-source data. The Lao administrative shapefile data was downloaded from the United Nations Office for the Coordination of Humanitarian Affairs (OCHA) website (https://data.humdata.org/dataset/cod-ab-lao), provided under free and open data (https://data.humdata.org/faqs/licenses); the topographic data of Earth's surface was downloaded from NASA Earthdata (https://www.earthdata.nasa.gov/data/instruments/srtm) and are made available under NASA's free and open data policy (https://www.earthdata.nasa.gov/engage/open-data-services-software/data-use-policy).

United Nations Office for the Coordination of Humanitarian Affairs (OCHA) website (https://data.humdata.org/dataset/cod-ab-lao), provided under free and open data (https://data.humdata.org/faqs/licenses); the topographic data of Earth's surface was downloaded from NASA Earthdata (https://www.earthdata.nasa.gov/data/instruments/srtm) and are made available under NASA's free and open data policy (https://www.earthdata.nasa.gov/engage/open-data-services-software/data-use-policy).

Site characteristics were classified as follows: (i) karstic cave/karstic mountains, (ii) rubber plantation, (iii) forest areas, and (iv) domestic animal sheds in rural villages in other provinces and urban villages in the capital of Vientiane (Table 1). Most collection sites were situated in karstic mountains, where villagers frequently utilize the caves for their livelihood, including hunting bats for food, collecting guano to fertilize their crops and, for some households, selling it for their economic benefit. In addition, people also cleared forested areas close to karstic mountains/caves for cattle farming and to grow crops. Each of these activities exposes Laotians to sandflies.

### Sandfly collection procedure

In each selected location, standard collection methods using CDC light traps (LT, John W. Hock Co., FL, U.S.A.) were used for adult sandfly collection for ~12-14h, between 4 pm and 9 am the following day. Sandflies were frozen at −20 °C for 20–30 minutes, and then sorted, labeled, and counted. All samples were stored in the field at −20 °C, and then transported to the IPL laboratory in Vientiane capital, where they were stored at −80 °C for further investigations.

**Table 1. Main characteristics of collection sites for sandflies in Laos.**

| Province | District | Main Coordinate | Altitude (in meters a.s.l.) | Month, Year of collection | Characteristic of sites |
|---|---|---|---|---|---|
| Bokeo | Huayxay | 20.4252, 100.3543 | 392 | May, 2017 | Domestic animal shed/Rural area |
| Luangnamtha | Long | 20.9657, 100.8414 | 550 | January, 2024 | Cave/Karstic area |
| | | | | | Rubber plantation |
| | Viengphoukha | 20.7238, 101.1533 | 730 | March and May, 2022 | Cave/Karstic area |
| Luangphabang | NamBak | 20.5851, 102.4524 | 370 | July, 2020 | Domestic animal shed/Rural area |
| | Ngoy | 20.5540, 102.6278 | 340 | July and September, 2020 | Cave/Karstic area |
| | | | | | Domestic animal shed/Rural area |
| | | | | | Rubber plantation |
| | Xiengern | 19.6766, 102.1104 | 400 | June, 2016 | Rubber plantation |
| Vientiane province | Feung | 18.5634, 101.9729 | 265 | November, 2014; February, and May, 2015; July, 2020; August and September, 2022 | Cave/Karstic area |
| | HinHeup | 18.7494, 102.2659 | 255 | May, 2016 | Cave/Karstic area |
| | Kasi | 19.1317, 102.1216 | 425 | June and July, 2022 | Cave/Karstic area |
| | | | | | Domestic animal shed/Rural area |
| | Vangvieng | 18.9755, 102.3228 | 329 | February, 2016 | Cave/Karstic area |
| Vientiane Capital | Hadsayfong, Naxaythong, Saysettha, Sisattanak, and Xaythany | 17.8243, 102.6539 | 170 | November, 2014; May and June, 2016 | Domestic animal shed/Urban area |
| Xayaboury | Boten | 17.6563, 101.1100 | 580 | April, 2017 | Domestic animal shed/Rural area |
| | | | | | Forest area |
| Xiengkhouang | Kham | 19.6743, 103.5682 | 616 | February and August 2020 | Cave/Karstic area |

## Preparation for sandfly identification and pathogen detection

To inventory sandfly species and their related pathogens, sandfly samples were randomly selected from different site characteristics and provinces for processing. In karstic areas where high numbers of sandflies were collected per trap, not all sandflies in the trap were processed. As a result, quantitative data analysis, such as the density of sandflies per trap-night could not be conducted in this study. For samples selected for processing, the head, wings, and abdomen genitalia of both sexes were cut under a stereomicroscope using sterile needles. The head, wings, and genitalia were mounted on slides using polyvinyl alcohol (PVA) mounting medium for sand fly identification. The thoraxes and abdomens were stored individually at -80°C for molecular analysis.

## Sandfly morphological identification

External and internal morphological characters were used to identify the sandfly at the genus and species levels. All slides were morphologically identified using the dichotomous keys of Lewis and other related original description references [15,18,20–28]. Regarding morphological classification, all specimens with poor-quality slide mounting were classified at the genus level by adding species (spp.) only. Due to morphologic similarity within the genus, *Sergentomyia* males were only identified at the genus level. Specimens that could not be classified as any species described in the literature, but whose main morphological characters were not complete such as broken palps or antennae flagellomeres etc. (as samples were stored in dry ice for pathogen screening) were classified as species (sp.) suffix by their closest related species and add "-like" or number.

## Nucleic acid extraction

Samples were homogenized for 2–5 min by a TissueLyser II system (Qiagen) with 0.6 mL of 1X phosphate-buffered saline (PBS) and Lysing Matrix E zirconium beads (MP Biomedicals). Nucleic acid was extracted using NucleoSpin8 extraction kit (MACHEREY-NAGEL GmbH & Co, Germany) by following the manufacturer's protocol.

## Sandfly cytochrome b gene database

To construct our genetic database for sandfly identification, the cytochrome b (*cyt-b*) was selected for amplification and sequencing in this study. The primers C3B-PDR and N1N-PDR [29] were used (S1 Table). Amplification was carried out in a 50 μL volume containing 5 μL of extracted DNA and 45 μL of PCR Master Mix (Promega) containing 50 pmol of each primer. The amplification conditions were followed as previously described [29]. Sequencing reactions were performed using the BigDye Terminator v1.1 cycle sequencing kit (Applied Biosystems) for Sanger sequencing (Applied Biosystems 3500xL Genetic Analyzer). Sequences of *Cyt-b* obtained from this study were deposited in GenBank under accession numbers PV054613– PV054752.

## Bloodmeal analysis

Semi-gravid sandfly samples were used to determine the bloodmeal origin. The vertebrate prepronociceptin (PNOC) and *cyt-b* genes were selected. Primers: PNOC(F) and PNOC(R) were used for amplifying vertebrates' PNOC gene. Universal vertebrate primers: Cytbvert2D (F) and Cytbvert1D (R) were also used for amplifying vertebrates' *cyt-b* gene (S1 Table). The amplification reaction and conditions were followed as previously described by Haouas et al. and Hadj-Henni et al. [30,31], DNA was extracted as previously described [32], and used as positive controls.

## Molecular detection of *Leishmania* and *Trypanosoma* species in sandflies

A nested PCR was performed on extracted DNA primarily to screen for *Leishmania* genetic material at the Lao-Oxford-Mahosot Hospital Wellcome Trust Research Unit (LOMWRU) in Vientiane, Laos. The PCR amplified the internal transcribed spacer 1 (ITS1 and 5.8S) region of the rRNA gene using primers LITSR and L5.8S for the primary PCR reaction and LITS2R and L5.8S inner for the secondary PCR reaction [33,34] (S1 Table). The PCR targeted the 300–350 bp fragment of the ITS1 + 5.8S region. Two positive controls (*L. major* and *L. tropica*) [35] and a negative control were used for every PCR run. Though these primer sets were designed for the detection of *Leishmania* sp., previous reports have shown that *Leishmania* primer sets amplifying the ITS1 + 5.8S region can also detect some *Trypanosoma* sp. PCR-positive products were purified using GeneJET Gel Extraction Kit (Thermofisher, UK) and sent for sequencing at a commercial facility (MACROGEN, South Korea) and also sequenced at IPL using the BigDye Terminator v1.1 cycle sequencing kit (Applied Biosystems) for Sanger sequencing (Applied Biosystems 3500xL Genetic Analyzer). Sequences of *Trypanosoma* spp. detected from this study were deposited in the GenBank database (PV034524 - PV034547).

## Phylogenetic analysis of sand fly species and detected parasites

Assembly sequences from sandflies and parasites were compared to the GenBank database using the BLASTn tool. For sandfly *cyt-b*, sequences were aligned using MAFFT v7.505 (with parameters: --maxiterate 1000 --localpair) [36], with pairwise distances calculated in MEGA 12 [37]. For *Trypanosoma* ITS1 + 5.8S sequences were aligned using MAFFT v7.505 (with parameters: --maxiterate 1000 --localpair; --multipair --addfragments) (Fig A-C in S1 Text) [36]. Alignments were manually curated in Geneious Prime v2025.1.2, where sequence ends with low-quality bases or unalignable regions were trimmed to ensure reliable homology, particularly for the hypervariable ITS1 region. Alignments were then cross-validated with MUSCLE and CLUSTALW (Fig D-E in S1 Text).

All curated alignments were further trimmed with BMGE v1.12 with default parameters [38]. The pairwise distance of sandfly nucleotide sequences was calculated in MEGA12 software [37]. Maximum likelihood phylogenetic analysis was carried out with IQ-TREE on the IQ-TREE web server (http://iqtree.cibiv.univie.ac.at/) with parameters: -m TEST -bb 1000 -alrt 1000 -abayes. The best-fit nucleotide substitution model was auto-selected based on Bayesian Information Criterion (BIC) [39] (Further details on the technical analysis of the *Trypanosoma* phylogenetic tree are provided in S1 Text). Finally, FigTree version 1.4.4 was used for phylogenetic tree viewing and editing.

## Results

### Sandfly species diversity

A total of 3,857 sandflies, of which 3,349 were females and 508 males, were morphologically identified. Sandflies were classified into 25 species belonging to five genera: *Chinius* (1 species): *Ch. eunicegalatiae*; *Idiophlebotomus* (1 species): *Id. longiforceps*, *Phlebotomus* (9 species): *Ph. barguesae*, *Ph. breyi*, *Ph.* (*Adlerius*) sp.1, *Ph. kiangsuensis*, *Ph. mascomai*, *Ph. argentipes*-like (*Ph. seowpohi*), *Ph. shadenae*, *Ph. sinxayarami*, *Ph. stantoni*, *Ph.* spp.; *Sergentomyia* (14 species): *Se. anodontis*, *Se. bailyi*, *Se. barraudi* group, *Se. brevicaulis*, *Se. dvoraki*, *Se. hivernus*, *Se. khawi*, *Se. perturbans*, *Se. phasukae*, *Se. sutherlandi* n. sp., *Se. sylvatica*, *Se. tambori*, *Se. gemmea*-like, *Se.* spp.; and *Grassomyia* (1 species): *Gr. indica* (Table 2).

Regarding species richness differences between provinces, the most diversity was found from provinces where sandfly collections were made from karstic limestone areas including Luangnamtha (Species richness (r) = 17 species), Luangphabang (r = 15), and Vientiane province (r = 21), except for Xiengkhouang (r = 6) where there was low diversity found from karstic areas. The lowest diversity was found from Bokeo (r = 8), Vientiane Capital (r = 6), and Xayaboury (r = 4) where sandflies were collected from domestic animal sheds and green forest areas (Table 3).

### Construction of cyt-b database and phylogenetic analysis of sandflies

A total of 141/150 samples from this study had the *cyt-b* gene successfully sequenced, including 20 species belonging to five genera (S2 Table). Analysis of these sequences together with other species reference sequences from our previous studies [21,22] and from GenBank (222 sequences in total) (S2 Table) shows that the genetic tree *cyt-b* was correlated with the morphological identification, especially for group species (Figs 2, and S1 for a full tree). Three sequences of *Phlebotomus* species that could not be identified by morphology due to the low quality of slide mounting were identified as *Ph. kiangsuensis* (2) and *Ph. mascomai* (1). For *Sergentomyia*, two were *Se. brevicaulis*, three were *Se. sutherlandi* n. sp., two were *Se. tambori*, and one was *Se. khawi*. *Phlebotomus (Adlerius)* sp.1 was separated from known *Ph. chinensis* from China (HM747275, HM747274, and HM747273) with a distance value of 0.14 (14%). The mean pairwise distance between species groups ranges from 0.02 to 0.41 (2–41%), and within species, ranging from 0 to 5%, especially in the *Se. barraudi* group (5%) (S3 Table).

*Sergentomyia brevicaulis* group was morphologically classified based on its original description in having long antennae flagellomere 1 (f1) and extending beyond the proboscis tip with about 50 posterior comb-like teeth and one row of anterior teeth on cibarium [16]. However, we observed that among samples from the northern part of Laos, posterior comb-like teeth ranged from 47-64 teeth with one or two rows of anterior teeth of 17 – 26 per row (Fig 3). A detailed image of a *Se. brevicaulis* specimen from Laos is provided in Fig 4. Interestingly, when analyzing sequences of *Se. rudnicki* from our previous study in Laos [22] and *Se. brevicaulis* identified here, the mean pairwise distance within these two species was very low, 0.014 (1.4%) (S3 Table). After morphological analysis, *Se. maiae* (Fig 3D) specimens were different from *Se. sutherlandi* n. sp. (Fig 3E and 3G). Together with *cyt-b* analysis, the pairwise distance between the two species was 12% (S3 Table). Consequently, based on morphological and molecular variability, we described *Se. sutherlandi* n. sp. as a new species.

### Description of new taxa of sandfly

The consensual terminology Galati et al. 2017 [40] has been used in this description.

Description of the female of *Sergentomyia sutherlandi* n. sp. Vongphayloth, Randrianambinintsoa & Depaquit

**Table 2. Number of sandfly species by sex from seven provinces.**

| Species | Province | | | | | | | |
|---|---|---|---|---|---|---|---|---|
| | BK | LNT | LPB | VTC | VTP | XK | XYR | Total |
| | n (f/m) | n (f/m) | n (f/m) | n (f/m) | n (f/m) | n (f/m) | n (f/m) | n (f/m) |
| *Ch. eunicegalatiae* | | | | | 207 (171/36) | | | 207 (171/36) |
| **Total** | | | | | **207 (171/36)** | | | **207 (171/36)** |
| *Id. longiforceps* | | 14 (11/3) | 5 (5/0) | | 264 (217/47) | | | 283 (233/50) |
| **Total** | | **14 (11/3)** | **5 (5/0)** | | **264 (217/47)** | | | **283 (233/50)** |
| *Ph. barguesae* | | 3 (3/0) | 3 (3/0) | | 3 (3/0) | 5 (5/0) | | 14 (14/) |
| *Ph. breyi* | | | | | 104 (97/7) | | | 104 (97/7) |
| *Ph. (Adlerius)* sp.1 | | 2 (2/0) | | | | | | 2 (2/) |
| *Ph. kiangsuensis* | | 27 (26/1) | 1 (1/0) | | 34 (26/8) | 7 (7/0) | | 69 (60/9) |
| *Ph. mascomai* | 3 (3/0) | 8 (8/0) | 2 (2/0) | | 153 (134/19) | 6 (6/0) | | 172 (153/19) |
| *Ph. seowpohi* | 20 (20/0) | 10 (10/0) | 46 (46/0) | 1 (0/1) | 4 (4/0) | | | 81 (80/1) |
| *Ph. shadenae* | | 4 (4/0) | | | | | | 4 (4/) |
| *Ph. sinxayarami* | | 4 (4/0) | 2 (2/0) | | 506 (485/21) | | | 512 (491/21) |
| *Ph. stantoni* | 6 (6/0) | 45 (44/1) | 12 (12/0) | 2 (1/1) | 66 (53/13) | 7 (7/0) | 3 (0/3) | 141 (123/18) |
| *Ph.* spp. | 2 (2/0) | 5 (5/0) | 11 (11/0) | | 92 (78/14) | | | 110 (96/14) |
| **Total** | **31 (31/0)** | **108 (106/2)** | **77 (77/0)** | **3 (1/2)** | **962 (880/82)** | **25 (25/0)** | **3 (0/3)** | **1209 (1120/89)** |
| *Se. anodontis* | | 40 (40/0) | 145 (145/0) | | 49 (49/0) | 140 (140/0) | | 374 (374/) |
| *Se. bailyi* | | 1 (1/0) | 13 (13/0) | 5 (5/0) | 14 (14/0) | | | 33 (33/) |
| *Se. barraudi* group | 18 (18/0) | 4 (4/0) | 20 (20/0) | 12 (12/0) | 69 (69/0) | | 7 (7/0) | 130 (130/) |
| *Se. brevicaulis* | | 91 (91/0) | | | 97 (97/0) | | | 188 (188/) |
| *Se. dvoraki* | 2 (2/0) | 2 (2/0) | 60 (60/0) | | 103 (103/0) | | 2 (2/0) | 169 (169/) |
| *Se. gemmea*-like | | | 2 (2/0) | | | | | 2 (2/) |
| *Se. hivernus* | 12 (12/0) | | 3 (3/0) | 13 (13/0) | 55 (55/0) | | 1 (1/0) | 84 (84/) |
| *Se. khawi* | 33 (33/0) | | 237 (237/0) | 12 (12/0) | 171 (171/0) | 2 (2/0) | | 455 (455/) |
| *Se. sutherlandi* n. sp. | | 17 (17/0) | | | | | | 17 (17/) |
| *Se. perturbans* | | | | | 1 (1/0) | | | 1 (1/) |
| *Se. phasukae* | | 2 (2/0) | 6 (6/0) | | 3 (3/0) | | | 11 (11/) |
| *Se. sylvatica* | | | 1 (1/0) | | 1 (1/0) | | | 2 (2/) |
| *Se. tambori* | | 36 (36/0) | | | 4 (4/0) | | | 40 (40/) |
| *Se.* spp. | 46 (2/44) | 30 (0/30) | 56 (50/6) | 16 (0/16) | 345 (143/202) | 4 (2/2) | 33 (0/33) | 530 (197/333) |
| **Total** | **111 (67/44)** | **223 (193/30)** | **543 (537/6)** | **58 (42/16)** | **912 (710/202)** | **146 (144/2)** | **43 (10/33)** | **2036 (1703/333)** |
| *Gr. indica* | 17 (17/0) | | 7 (7/0) | | 92 (92/0) | | 6 (6/0) | 122 (122/) |
| **Total** | **17 (17/0)** | | **7 (7/0)** | | **92 (92/0)** | | **6 (6/0)** | **122 (122/)** |
| **Total** | **159 (115/44)** | **345 (310/35)** | **632 (626/6)** | **61 (43/18)** | **2437 (2070/367)** | **171 (169/2)** | **52 (16/36)** | **3857 (3349/508)** |
| Species richness* | 8 | 17 | 17 | 7 | 21 | 6 | 5 | 25 |

*Specimens classified as species (spp.) were not counted. *Ch.*:*Chinius*, *Ph.*: *Phlebotomus*, *Se.*: *Sergentomyia*, and *Gr.*: *Grassomyia*. BK: Bokeo, LNT: Luangnamtha, LPB: Luangphabang, VTP: Vientiane province, VTC: Vientiane Capital, XYR: Xayaboury, and XK: Xiengkhouang. n: number of sandflies, f: female, and m: male.

(Fig 5A–5H, Table 4)
Genus: *Sergentomyia*
Subgenus: ungrouped waiting a *Sergentomyia*'s taxonomic revision
Measurements indicated are those of the holotype (voucher SF22-EX651).

**Table 3. Number of sandfly species by the site characteristics from seven provinces.**

| Province | BK | LNT | | LPB | | | VTC | VTP | | XK | XYR | | |
|---|---|---|---|---|---|---|---|---|---|---|---|---|---|
| Site characteristic | DAS | CKA | RP | CKA | RP | DAS | DAS | CKA | DAS | CKA | FA | DAS | Total |
| **Species** | | | | | | | | | | | | | |
| *Ch. eunicegalatiae* | | | | | | | | 207 | | | | | 207 |
| **Total** | | | | | | | | **207** | | | | | **207** |
| *Id. longiforceps* | | 14 | | 5 | | | | 264 | | | | | 283 |
| **Total** | | **14** | | **5** | | | | **264** | | | | | **283** |
| *Ph. barguesae* | | 3 | | 3 | | | | 3 | | 5 | | | 14 |
| *Ph. breyi* | | | | | | | | 104 | | | | | 104 |
| *Ph. (Adlerius)* sp.1 | | 2 | | | | | | | | | | | 2 |
| *Ph. kiangsuensis* | | 27 | | 1 | | | | 34 | | 7 | | | 69 |
| *Ph. mascomai* | 3 | 8 | | 2 | | | | 153 | | 6 | | | 172 |
| *Ph. seowpohi* | 20 | 10 | | 46 | | | 1 | 4 | | | | | 81 |
| *Ph. shadenae* | | 4 | | | | | | | | | | | 4 |
| *Ph. sinxayarami* | | 4 | | 2 | | | | 506 | | | | | 512 |
| *Ph. stantoni* | 6 | 43 | 2 | 7 | | 5 | 2 | 59 | 7 | 7 | 2 | 1 | 141 |
| *Ph.* spp. | 2 | 5 | | 11 | | | | 89 | 3 | | | | 110 |
| **Total** | **31** | **106** | **2** | **72** | | **5** | **3** | **952** | **10** | **25** | **2** | **1** | **1209** |
| *Se. anodontis* | | 40 | | 144 | | 1 | | 49 | | 140 | | | 374 |
| *Se. bailyi* | | 1 | | 7 | 6 | | 5 | 14 | | | | | 33 |
| *Se. barraudi* group | 18 | 4 | | 18 | 2 | | 12 | 67 | 2 | | 7 | | 130 |
| *Se. brevicaulis* | | 91 | | | | | | 97 | | | | | 188 |
| *Se. dvoraki* | 2 | 2 | | 28 | 9 | 23 | | 97 | 6 | | | 2 | 169 |
| *Se. gemmea*-like | | | | | 2 | | | | | | | | 2 |
| *Se. hivernus* | 12 | | | 3 | | | 13 | 52 | 3 | | 1 | | 84 |
| *Se. khawi* | 33 | | | 163 | 73 | 1 | 12 | 171 | | 2 | | | 455 |
| *Se. sutherlandi* n. sp. | | 17 | | | | | | | | | | | 17 |
| *Se. perturbans* | | | | | | | | 1 | | | | | 1 |
| *Se. phasukae* | | 2 | | 6 | | | | 3 | | | | | 11 |
| *Se. sylvatica* | | | | 1 | | | | 1 | | | | | 2 |
| *Se. tambori* | | 36 | | | | | | 4 | | | | | 40 |
| *Se.* spp. | 46 | 30 | | 50 | 4 | 2 | 16 | 335 | 10 | 4 | 28 | 5 | 530 |
| **Total** | **111** | **223** | | **420** | **96** | **27** | **58** | **891** | **21** | **146** | **36** | **7** | **2036** |
| *Gr. indica* | 17 | | | | 4 | 3 | | 92 | | | 2 | 4 | 122 |
| **Total** | **17** | | | | **4** | **3** | | **92** | | | **2** | **4** | **122** |
| **Grand Total** | **159** | **343** | **2** | **497** | **100** | **35** | **61** | **2406** | **31** | **171** | **40** | **12** | **3857** |
| **Species richness*** | **8** | **17** | **1** | **15** | **6** | **5** | **6** | **21** | **4** | **6** | **4** | **3** | **25** |

*Specimens classified as species (spp.) were not counted. *Ch.*:*Chinius*, *Ph.*: *Phlebotomus*, *Se.*: *Sergentomyia*, and *Gr.*: *Grassomyia*. BK: Bokeo, LNT: Luangnamtha, LPB: Luangphabang, VTP: Vientiane province, VTC: Vientiane Capital, XYR: Xayaboury, and XK: Xiengkhouang. DAS: Domestic animal shed, CKA: Cave/Karstic area, RP: Rubber plantation, and FA: green forest area.

*Head*

Occiput with two narrow lines of well-individualized setae (Fig 5A).

Clypeus 131 µm long and 100 µm wide, with 38 setae randomly distributed.

Eyes 197 µm long, 113 µm wide, with about 80 facets.

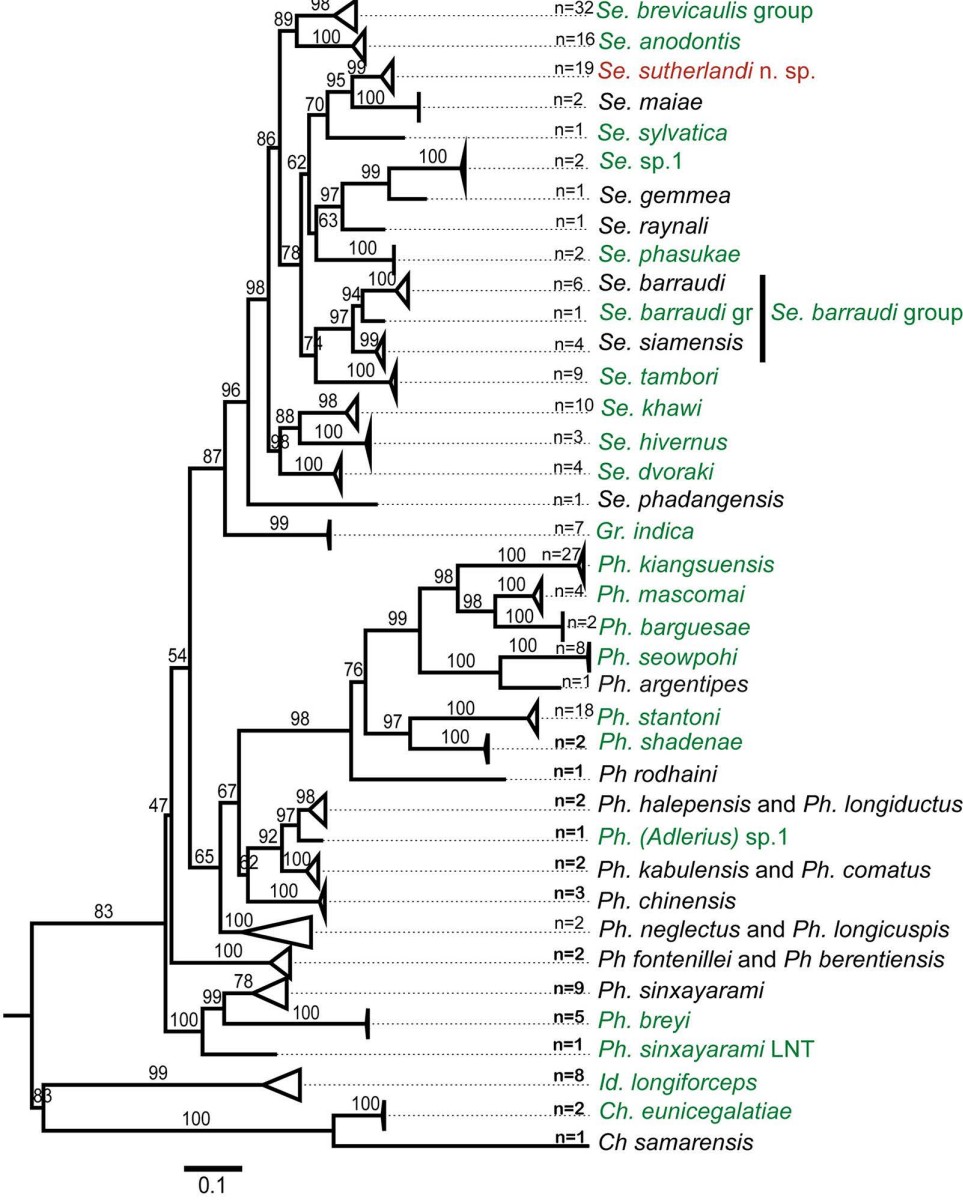

**Fig 2. Maximum likelihood phylogenetic tree constructed by IQ tree from 222 *cyt-b* sequences of 397 nt using a model that auto-selected based on Bayesian Information Criterion (BIC).** The numbers on the branches represent the bootstrap values (%) derived from 1,000 bootstrap replicates (options: -m TEST -bb 1000 -alrt 1000 -abayes). Reference sequences of each species generated in previous studies and available in GenBank were selected and included in this analysis. Green color indicates species found in this study. New species that are closely related to *Se. maiae* are labelled in red color. "n" indicates the number of sequences used for the analysis.

Interocular sutures incomplete. Interantennal sutures do not reach the interocular sutures.

Flagellomere 1 longer than f2 + f3 (Fig 5C).

Presence of 2 ascoids from f1 to f13. Absence on f14. Ascoidal formula: 2/f3- f13 with long ascoids, not reaching the next article.

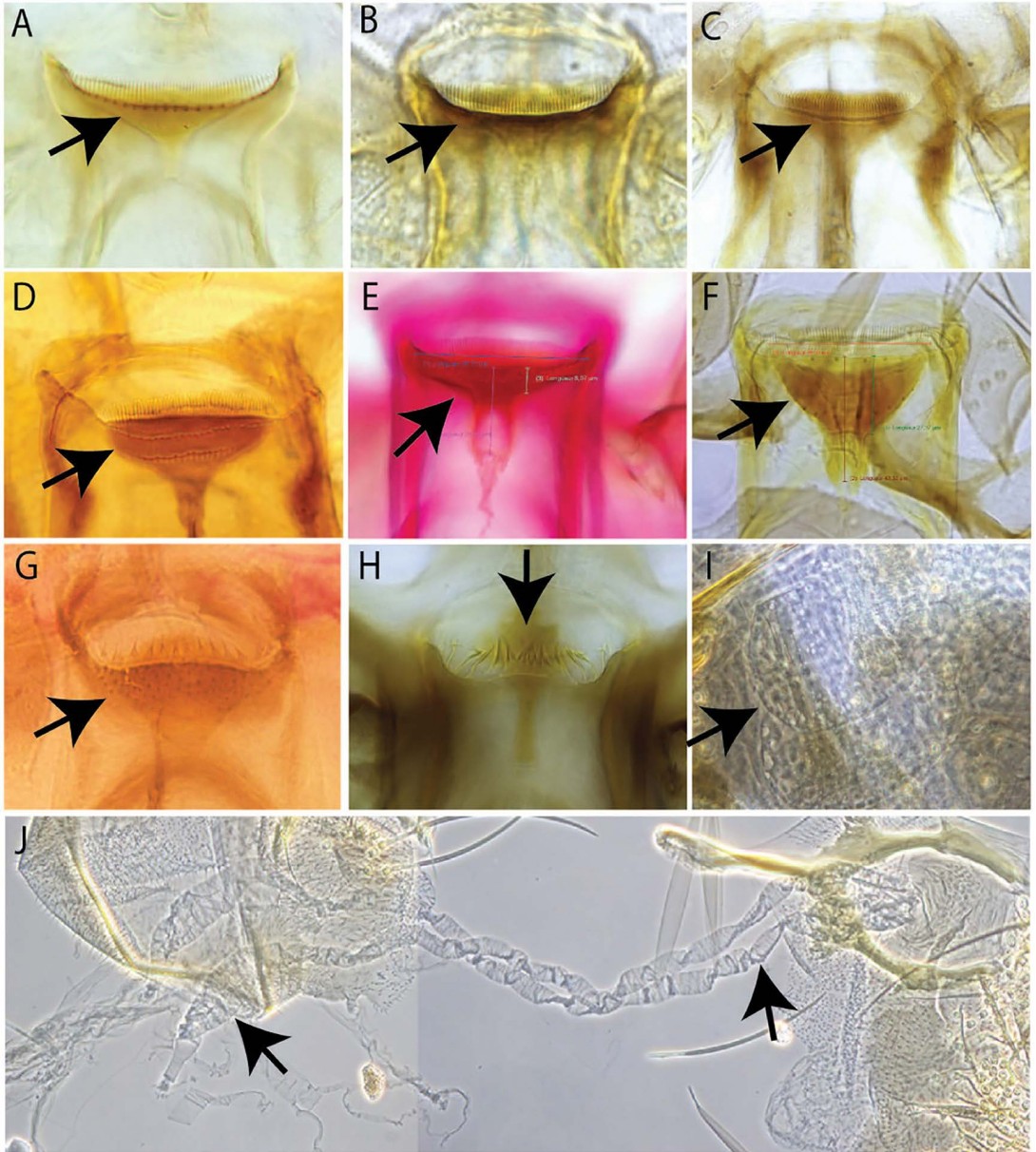

**Fig 3. Female morphological characteristics of sandflies.** *Sergentomyia barraudi* group **(A, B)**: A: cibarium of *Se. siamensis* with anterior teeth, B: cibarium of *Se. barraudi* group without anterior teeth; *Se. brevicaulis* group **(C, D)**: C: cibarium with one row of anterior teeth, and D: cibarium with two rows of anterior teeth; E: cibarium of *Se. maiae* paratype from Thailand; F: cibarium of *Se. sutherlandi* n. sp.; G: cibarium of *Se. gemmea*-like; *Se. tambori* (H, **I**): H: cibarium and I: spermathecae; and J: spermatheca of *Ph.* (*Adlerius*) sp. 1 (photo not to scale).

One papilla on f1 and f2. Absence of papilla from f3 to f6, two papillae on f7 and f8, three on f9, four on f10 and 11, and six on f12 to f14.

No simple setae on f1 to f5, two on f6, three f7, four on f8 and f9; three on f10, five on f11 and f12, 8 – 12 on f14.

Palpal formula: 1, 2, 3, 4, 5 (Fig 5D). Presence of a group of about 40 club-like Newstead's sensillae implanted proximally on the third palpal segment 3 (Fig 5E). Presence of one distal simple seta on p3, 9 on p4, and more than 10 on p5.

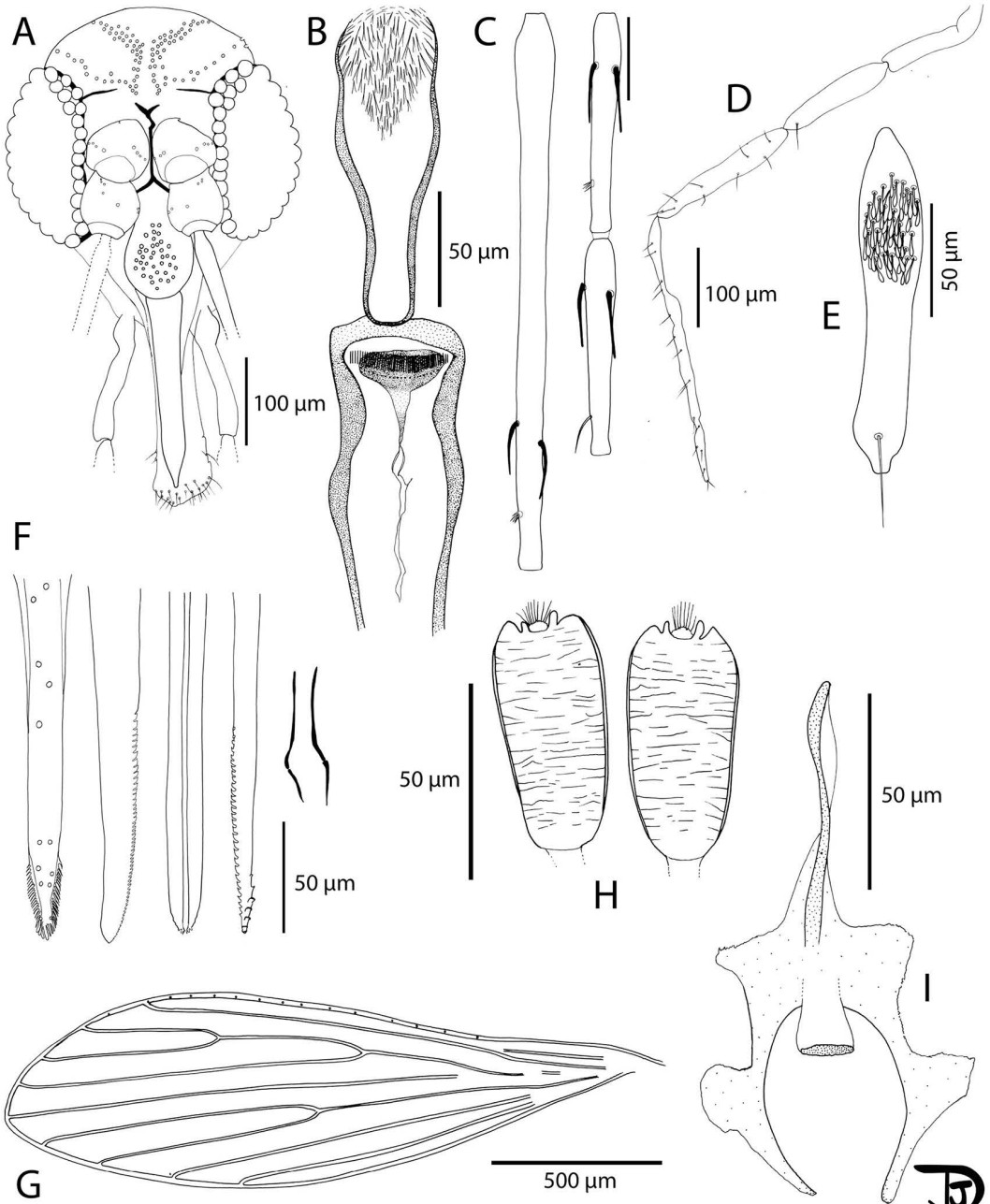

**Fig 4. *Sergentomyia brevicaulis* drawing from a Lao female.** A: head (TM7-SM-LNT-76); B: pharynx and cibarium (TM7-SM-LNT-38).; C: flagellomeres 1, 2 and 3 (= AIII, AIV and AV) (TM7-SM-LNT-80); D: palp (TM7-SM-LNT-80); E: third segment of the palp (P3) (TM7-SM-LNT-507); F: mouth parts (labrum-epipharynx, mandible, hypopharynx, maxilla, and labial furca from left to right) (TM7-SM-LNT-506); G: wing (TM7-SM-LNT-38); H: spermathecae (TM7-SM-LNT-506) and I: furca (TM7-SM-LNT-83).

Labrum 207 µm long.

Hypopharynx with faint ridges, groups of spiculated ridges on either side of the salivary canal. Maxillary lacinia exhibits nine external and about thirty teeth. Labial furca opened (Fig 5F).

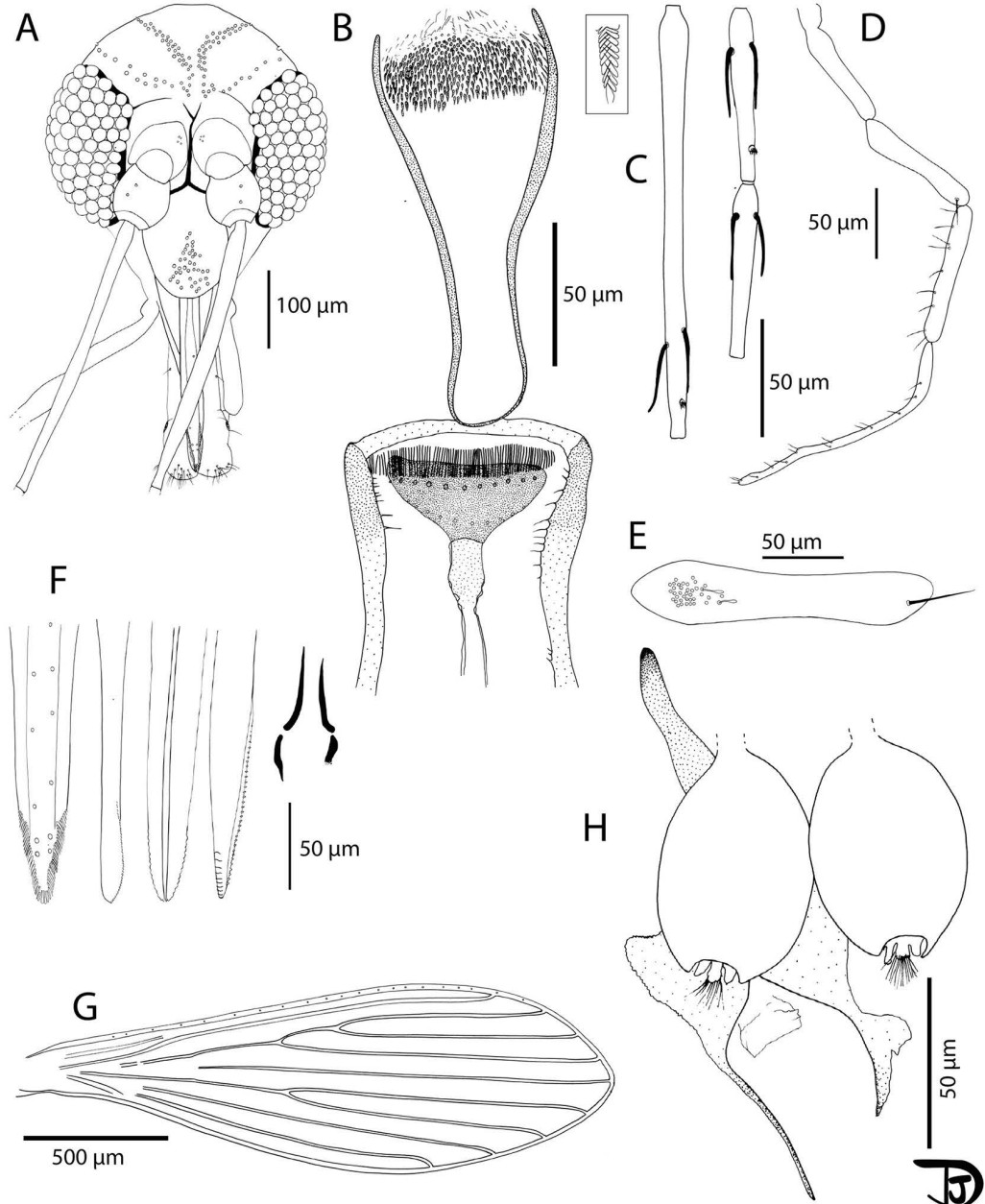

**Fig 5. *Sergentomyia sutherlandi* n. sp. Female.** A: head (holotype SF22-EX651); B: pharynx and cibarium (holotype SF22-EX651) with a view showing the implantation of the pharyngeal teeth in the rectangular box; C: flagellomeres 1, 2 and 3 (= AIII, AIV and AV)(paratype EX22-EX677); D: palp (paratype EX22-EX650); E: third segment of the palp (P3)(paratype EX22-EX650); F: mouth parts (labrum-epipharynx, mandible, hypopharynx, maxilla, and labial furca from left to right)(paratype EX22-EX695); G: wing (paratype EX22-EX650); H: furca and spermathecae (paratype EX22-EX695).

Comb-like cibarium armature with 77 palisadic posterior teeth. Presence of anterior teeth in two rows, ten denticles on each row. Sclerotized area with a broad fore end (Fig 5B).

*Cervix*

Two cervical sensillae on each side.

**Table 4. Comparative measurements (in μm) and counts between *Se. sutherlandi* n. sp. and *Se. maiae* females.**

| Variables | | *Se. sutherlandi* n. sp. | | *Se. maiae* | |
|---|---|---|---|---|---|
| | | No. of sample | Mean ± SD (Min-Max) | No. of sample | Mean ± SD (Min-Max) |
| Clypeus (length) | | 9 | 124.04 ± 12.16 (100 - 137) | 12 | 150.3 ± 19 (100.45 - 166.73) |
| Labrum (length) | | 9 | 205.49 ± 11.97 (187 - 223) | 12 | 213.12 ± 18.21 (184.37 - 244.52) |
| Flagellomeres | f1 | 10 | 335.1 ± 18.33 (312.86 - 365) | 12 | 372.08 ± 27.8 (330.08 - 418.38) |
| | f2 | 10 | 135.72 ± 7.81 (123.59 - 147) | 12 | 142.95 ± 9.61 (129.11 - 161.1) |
| | f3 | 10 | 140.13 ± 5.81 (133 - 150) | 12 | 148.91 ± 9.74 (132.57 - 163.35) |
| | f2 + f3 | 10 | 275.85 ± 13.15 (258.19 - 295) | 12 | 291.86 ± 19.35 (261.68 - 324.45) |
| | f1/f2 + f3 | 10 | 1.21 ± 0.04 (1.17 - 1.3) | 12 | 1.27 ± 1.44 (1.26 - 1.29) |
| Palpi | p1 | 12 | 37.35 ± 5.99 (29 - 47) | 12 | 36.43 ± 4.63 (30.98 - 43.68) |
| | p2 | 12 | 116.51 ± 6.93 (105 - 125) | 12 | 125.51 ± 10.43 (107.14 - 139.25) |
| | p3 | 12 | 170.21 ± 6.33 (156 - 179) | 12 | 178.88 ± 15.76 (150.49 - 202.47) |
| | p4 | 12 | 195.96 ± 11.13 (176 - 212) | 12 | 209.26 ± 21.9 (177.45 - 252.58) |
| | p5 | 9 | 324.4 ± 46.82 (224 - 385) | 12 | 362.57 ± 36.14 (306 - 413.37) |
| Cibarium | Posterior teeth | 4 | 76.5 ± 2.38 (73 - 78) | 12 | 62.5 ± 5.5 (57 - 69) |
| | Anterior teeth row1 | 10 | 13.5 ± 0.97 (10 - 14) | 12 | 9.4 ± 2.07 (8 - 13) |
| | Anterior teeth row2 | 7 | 11.29 ± 0.95 (10 - 12) | 6 | 9.6 ± 1.1 (9 - 12) |
| Sclerotized area | Width | 13 | 56.48 ± 2.29 (53 - 59) | 6 | 49.2 ± 7.18 (37 - 55) |
| | Height to top | 13 | 45.83 ± 2.15 (41.8 - 49) | 6 | 27.2 ± 3.61 (24 - 34) |
| | Base | 13 | 22 ± 2.86 (16 - 26) | 6 | 10.2 ± 2.76 (8 - 15) |
| Wing | Length | 3 | 1954.38 ± 42.35 (1908 - 1991) | 12 | 1910.3 ± 204.41 (1549.62 - 2203.24) |
| | Width | 3 | 615.6 ± 16.57 (593.58 - 632.21) | 12 | 581.86 ± 107.3 (349.69 - 725.87) |
| | α (r2) | 3 | 682.17 ± 19.21 (655.5 - 700) | 12 | 618.48 ± 170.91 (217.45 - 745.7) |
| | ε (r3) | 3 | 800.98 ± 30.09 (759 - 828) | 12 | 740.42 ± 164.26 (338.48 - 886.07) |
| | θ (r4) | 3 | 1074.17 ± 30 (1033.3 - 1104) | 12 | 1051.56 ± 173.51 (634.68 - 1214.51) |
| | r5 | 3 | 1384.01 ± 33.26 (1345.5 - 1425) | 12 | 1346.3 ± 205.44 (814.19 - 1547.09) |
| | β (r2 + r3) | 3 | 200.36 ± 9.66 (189.07 - 209) | 12 | 248.31 ± 17.93 (213.02 - 270.21) |
| | δ (r2+3 to r1) | 3 | 514.92 ± 22.04 (484.5 - 534) | 12 | 445.48 ± 152.52 (88.37 - 559.54) |
| | γ (r2+3+4) | 3 | 260.87 ± 38.01 (209 - 287) | 12 | 289.05 ± 49.64 (163.05 - 347.95) |
| | π (r2+3 to m1+2) | 3 | 33.67 ± 13.85 (26 - 40) | 12 | 39.3 ± 7.33 (30.04 - 49.41) |
| | α/β (r2/r2+3) | 3 | 3.41 ± 0.25 (3.23 - 3.7) | 12 | 2.49 ± 9.53 (1.02 - 2.76) |

One ventro-cervical sensillae was observed.

*Thorax and legs*

No thorax or legs observable on the sample taking into account they have been cut for pathogen screening.

*Wings*

Length = 1964 μm; Width = 632 μm. (Fig 5G)

r5 = 1381 μm, α (r2) = 700 μm, β (r2+3) = 189 μm, δ (r2+3-r1) = 526 μm, γ (r2+3+4) = 287 μm, ε (r3) = 815 μm, θ (r4) = 1085 μm, π = 35 μm. α (r2)/ β (r2+3) = 3.7.

*Abdomen*

Setae randomly distributed from segment V to X (no abdomen segment I-IV).

*Genitalia*

Smooth elongated capsule-like spermathecae. Walls and short terminal knob embedded in the capsule. Ducts impossible to observe. Probable basal common duct. Genital furca with two well-developed lateral processes (Fig 5H).

*Derivatio nominis*. The species *Sergentomyia sutherlandi* n. sp. is dedicated to our colleague Ian Sutherland, whose initiative contributed to the vector-borne pathogen surveillance program of the medical entomology laboratory at the Institut Pasteur du Laos.

The type locality of *Sergentomyia sutherlandi* n. sp. is Kao Lao cave (20.7238N, 101.1533E), Viengphukha district, Luangnamtha province, Laos.

**Type specimens.** The holotype female (voucher SF22-EX651) of *Sergentomyia sutherlandi* n. sp. and one female paratype (voucher SF22-EX695) have been deposited at the Terrestrial Arthropods Collection of the Muséum national d'Histoire naturelle (MNHN, Paris) under the inventory numbers MNHN-ED-11547 and MNHN-ED-11548.

To meet the criteria of availability, the authors Vongphayloth, Randrianambinintsoa & Depaquit are responsible for the name *Sergentomyia sutherlandi* n. sp. and should be cited as the sole authority of this taxon, according to Article 50 (1) of the International Code of Zoological Nomenclature.

### *Leishmania* detection

DNA was extracted from 1,464 sandflies, and nested PCR primarily for *Leishmania* sp. detection was performed on 227 pools, which included 20 sandfly species from six provinces (S4 Table).

Three pools were positive by nested-PCR. The individual samples from these pools were screened using the same PCR, with 24 individual samples positive. ITS1 + 5.8S sequences obtained were compared with sequences from GenBank. BLAST analysis showed that none of the sequences corresponded to *Leishmania*, but did, however, match with sequences of *Trypanosoma* spp. Among the 24 individual sandflies positive for *Trypanosoma* spp., two were *Id. longiforceps* collected from Kasi district, Vientiane province (1) and Viengphouka district, Luangnamtha province (1) in 2022, and 22 were *Ch. eunicegalatiae* collected in the same CDC trap from Feung district, Vientiane province in 2022.

Maximum likelihood phylogenetic analysis of partial ITS1 + 5.8S rRNA sequences (24 from Laotian sandflies, 53 from GenBank; Table A in S1 Text) showed high sequence divergence across MAFFT, MUSCLE, and CLUSTALW alignments (Fig C–E in S1 Text). In the primary tree (MAFFT-aligned, BMGE-trimmed, IQ Tree server, K2P+G4 model auto-selected with parameters: -m TEST -bb 1000 -alrt 1000 -abayes; Fig 6), the 24 sequences formed a monophyletic clade, but only with 50% bootstrap support. Within this clade, 22 sequences from *Chinius eunicegalatiae* clustered with *Trypanosoma* sp. (OL332789) from *Sergentomyia* sp. (Satun, Thailand) [41], with low bootstrap support (58%). Two sequences from *Idiophlebotomus longiforceps* (SF22_Ex243, SF22_Ex278) grouped with *Trypanosoma* sp. (OL332785) from *Sergentomyia khawi* (Songkhla, Thailand) [38], with only 75% bootstrap support. The clade was phylogenetically distinct from known *Trypanosoma cruzi* (e.g., AF362821, FJ555667, AY230234), *Trypanosoma lewisi* (e.g., EU861192), and *Trypanosoma theileri* (e.g., JN673395) clades. Phylogenetic trees constructed using other alignment algorithms, e.g., MUSCLE, CLUSTALW, or MAFFT (Fig F–H in S1 Text), showed varying topologies with low to moderate bootstrap support, including shifts in sister-group placement for some sequences compared to Fig 6.

### Blood meal detection

A total of 29 semi-gravid females or samples suspected of having a recent blood meal were processed to determine the blood meal origin. Unfortunately, except for positive controls, no sample was successfully amplified, which is likely due to over-digestion of the blood meals.

### Discussion

This is the first large-scale reporting of sandfly species diversity from seven provinces of northern Laos and the first attempt to detect *Leishmania* from sandflies in the country. There were 25 species identified, belonging to five genera, including two novel species. While no *Leishmania* DNA was detected from the screened sandflies, *Trypanosoma* DNA was detected in 24 samples using the same PCR method.

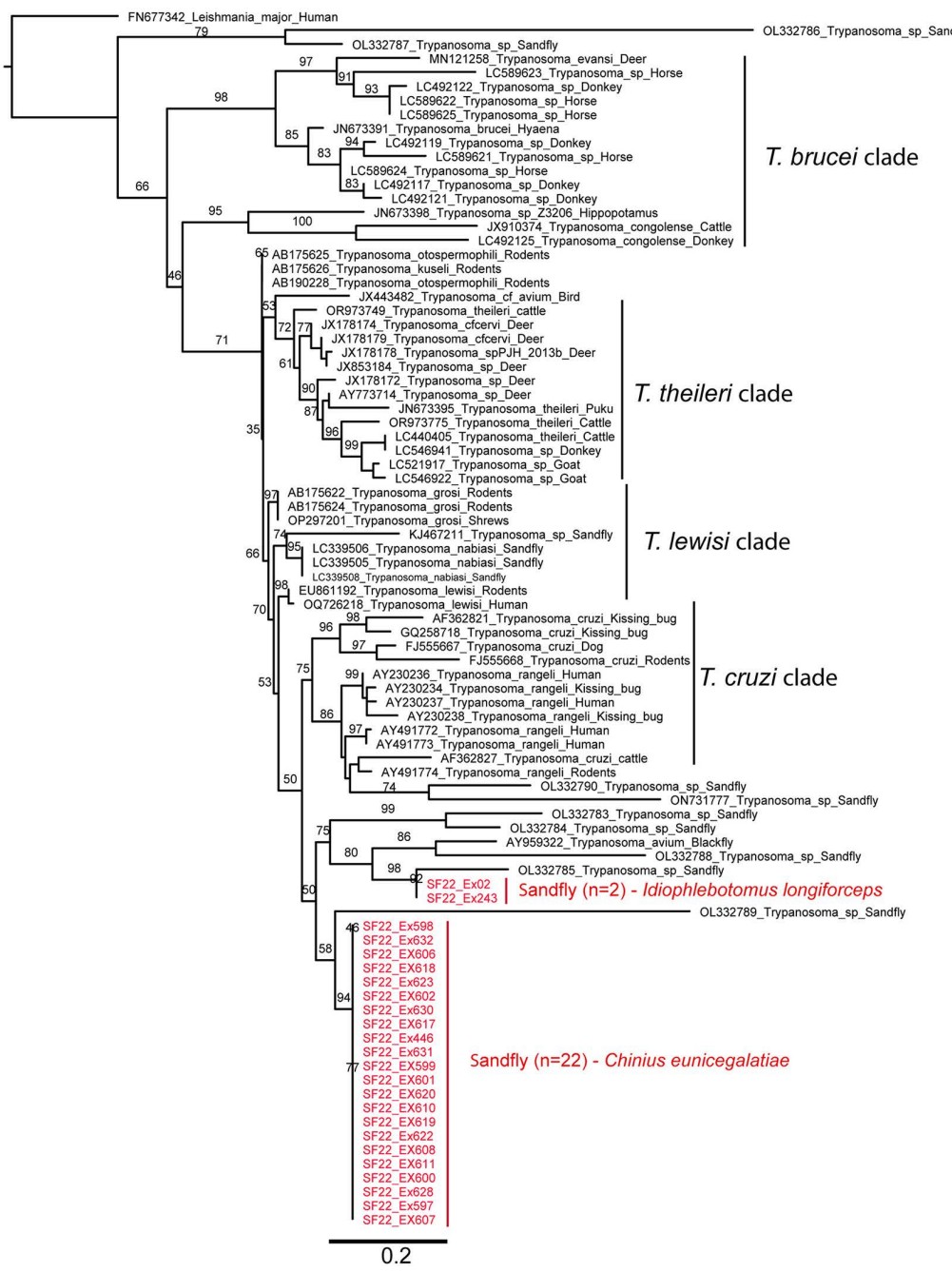

**Fig 6. Maximum likelihood phylogenetic tree constructed from *Trypanosoma* partial ITS1 and 5.8S region.** Sequences from BLAST results were selected and included for analysis. The tree was constructed using IQ-TREE using the auto-selected model (K2P+G4) based on Bayesian Information Criterion (BIC) with parameters -m TEST -bb 1000 -alrt 1000 -abayes. The numbers on the branches represent the bootstrap values (%) derived from 1000 replicates. Sequences obtained from this study in Laos are highlighted in red. *Leishmania major* (FN677342) was used as the outgroup.

A total of 25 sandfly species of five genera were identified. The greatest diversity was found in provinces where sandfly collections were made in karstic limestone areas, including Luangnamtha, Luangphabang, and Vientiane provinces. The lowest diversity was identified in Bokeo, Vientiane Capital, and Xayaboury provinces, where sandflies were collected from domestic animal sheds and green forest areas. Such heterogeneous species diversity and composition of sandflies according to trapping habitats have been observed in countries neighboring Laos, including Vietnam and Thailand and in other parts of the world, such as Africa and South America [11–14,42–44]. Among sandfly species found, *Ph. stantoni*, *Ph. argentipes*-like, *Se. barraudi* group, *Se. dvoraki, Se. hivernus,* and *Se. khawi* are widely distributed across different landscapes from karstic areas to domestic animal areas as well as in urban areas in Vientiane capital. This indicates that these species are well adapted for different environments.

**Sandfly taxonomic comments**

**Genus *Chinius*.** ***Chinius eunicegalatiae*** was found only in Vientiane province, where it was originally described from a cave in Vangvieng district [20]. From our previous results [22] and this study, this species seems to be restricted to karstic areas of Vientiane province. The ecology of this species is still not well known, *Trypanosoma* DNA was detected from *Ch. eunicegalatiae* in this study.

**Genus *Idiophlebotomus*.** ***Idiophlebotomus longiforceps*** had been recorded in Vientiane province from our previous study [22]. In the present study, this species was found widely distributed in karstic areas from Luangnamtha, Luangphabang and Vientiane provinces. Karstic areas, with their humid microclimates, abundant sheltered crevices, organic-rich soils, and presence of vertebrate hosts, may create ideal conditions for this sandfly species.

**Genus *Phlebotomus*.** A total of eight species have been recorded in Laos before this report, including *Ph. argentipes, Ph. barguesae*, *Ph. breyi*, *Ph. mascomai*, *Ph. kiangsuensis*, *Ph. shadenae*, *Ph. sinxayarami,* and *Ph. stantoni* [18,21,22,25]. In the present study, we recorded a new *Phlebotomus* subgenus (*Adlerius*) and a likely new species in Laos. Excluding China, no *Adlerius* species have been reported in the region [11–14,24,44]. In the present study, two females were collected from Luangnamtha province, located in the northern part of Laos bordering Yunnan province of China, where three species of subgenus *Adlerius* are known to occur including *Ph. chinensis, Ph. fengi,* and *Ph. sichuanensis* [45]. The taxonomy of the subgenus *Adlerius* is known to be one of the most difficult to identify morphologically at the species level within the *Phlebotominae* due to the characteristic similarities among females within this subgenus [46]. The spermathecae are characteristic and exclusive of the subgenus. However, most of the females are indistinguishable, and their identification at the species level remains impossible. The identification is based on molecular markers and male morphology, commonly focusing on genitalia and antennal formula [47,48]. Despite the genetic variability of female specimens from Laos, we cannot describe a new species due to the lack of associated males. Further captures of both males and females are required to conclude definitively about the taxonomic status of this population of *Adlerius* from Laos. *Phlebotomus argentipes* was recorded in Laos from Luangphang province [18]. However, the records of *Ph. argentipes* in Laos as well as in other Southeast Asian countries including their morphology and genetic descriptions, should be investigated in the light of the results of our study. *Phlebotomus argentipes* was first described from India, and its morphological and biological varieties according to geographical distribution have been discussed by Lewis [24]. Subsequently, another closely related species, *Ph. mascomai*, had been described from Thailand [27]. In the present study, we didn't record any *Ph. argentipes* but only specimens we identified as *Ph. argentipes*-like or *Ph. seowpohi* Depaquit, Vongphayloth & Tan, 2025 [49]. Despite a morphological resemblance to *Ph. argentipes*, the specimens from this study exhibit important genetic divergence (Fig 2). This species has been recently described from Singapore and could be more widely distributed in Southeast Asia as previously expected.

Our *cyt-b* phylogeny is congruent with earlier work [29], which demonstrated the value of mitochondrial haplotype analysis for resolving taxonomic relationships and population structure in *Phlebotomus* subgenera such as *Phlebotomus* and *Larroussius*. This supports its broader applicability for distinguishing closely related species, including *Adlerius* spp. [48].

**Genus *Sergentomyia*.** In Laos and adjacent countries, the taxonomy of the genus *Sergentomyia* remains unclear and an in-depth taxonomic revision as previously described on sandflies in Laos, Thailand, and Vietnam is needed [22,44,50]. Among sandfly species in Southeast Asia, the genus *Sergentomyia* is the most morphologically and genetically diverse. More studies have been conducted, and new species described [14,22,25,51]. Here we describe an additional new species of sandflies from Laos.

*Sergentomyia sutherlandi* n. sp. was closely related to *Se. maiae* and *Se. mahadevani*, two species originally described from Thailand [24,25]. *Se. sutherlandi* morphologically contrasted easily with *Se. maiae* by: (1) number of posterior teeth on cibarium, which ranges from 73-78 in *Se. sutherlandi* n. sp. *vs* 57–69 in *Se. maiae*; (2) Thick sclerotized area with a broad fore end in *Se. sutherlandi* n. sp. (Fig 3D) *vs* a thin sclerotized area with a narrow fore end in *Se. maiae* (Fig 3F). *Sergentomyia sutherlandi* n. sp. can be separated from *Se. mahadevani* by (1) more cibarial posterior teeth in *Se. sutherlandi* n. sp. (73–78) than *Se. mahadevani* (about 50) [24]; (2) both species have sclerotized area with a broad fore end but *Se. sutherlandi* n. sp. has two rows of anterior teeth (Fig 3D) *vs* no anterior tooth in *Se. mahadevani*.

*Sergentomyia brevicaulis* is closely related to *Se. rudnicki* and *Se. barraudi* group in presenting a comb-like cibarial armature. *Sergentomyia barraudi* group could be separated from *Se. rudnicki* and *Se. brevicaulis* by their shorter antennae flagellomere 1, and cibarial sclerotized area forked anteriorly. *Sergentomyia brevicaulis* was originally described from Vietnam in 1962 in having cibarium with about 50 comb-liked teeth and one row of anterior teeth [18]. Later, *Se. rudnicki* was described from specimens from Malaysia in 1978 by females with cibarium exhibiting about 90-comb-like teeth, two rows of vertical teeth about 22, pharyngeal teeth long, and oblong and annulate spermatheca [24]. In the present study, we observed that cibarium teeth of *Se. brevicaulis* varied in number of posterior teeth and row of anterior teeth; genetically *cyt-b* showed more than 98% similarity, indicating these two species of specimens from Laos could be varieties of each other. To confirm this observation, further taxonomic studies on a larger scale among Southeast Asian countries comparing both morphology and genetics of Laotian specimens with *Se. brevicaulis* from Vietnam and *Se. rudnicki* from Malaysia, where the original samples were described, is needed.

*Sergentomyia anodontis* s.l. was first recorded in Laos and discussed in our previous study [22]. Future larger-scale on morphological and molecular studies are clearly needed to further clarify the taxonomy of this probable complex of species. The present study showed that this species is widely distributed and restricted to karstic areas in Laos, including Luangnamtha, Luangphabang, Vientiane province and Xiangkhoung. No sample of this species was collected from other habitats except one specimen collected from domestic animal shed located near to a karstic cave in Luangphabang province.

*Sergentomyia bailyi* s.l. was recorded in Laos by Quate from burn tree holes [18]. In the present study we found this species in a variety of habitats, including domestic animal sheds in Vientiane capital, rubber plantations, and karstic areas. As already discussed by Vu et al. [44], this species may represent a species group; a taxonomic revision of this species is needed by studying more samples and genetic studies in Southeast Asian countries.

*Sergentomyia iyengari* group is considered one of the most taxonomically confusing among *Sergentomyia* genus in Southeast Asia. However, after more recent studies, species reported in this region may include *Se. hivernus*, *Se. dvoraki*, *Se. khawi* and *Se. gemmea* [22]. However, the records of *Se. gemmea* in Laos are pending revision, as the pairwise distance of the *cyt-b* gene between specimens from Laos and Thailand was high, 0.06 (6%) [15]. In this study, we considered this species as *Se. gemmea*-like (*Se*. sp 1). We did not consider the specimens from this study to exhibit all the characteristics necessary for a new species. Instead, we will await specimens exhibiting all the appropriately unique morphologic characteristics (antennae, palps, wings).

*Sergentomyia perturbans* and *Se. sylvatica* were also recorded in Laos by Quate in Vientiane from tree holes [18] and also in karstic areas in Vientiane province [22]. In this study, *Se. perturbans* was found only in Vientiane province, whereas *Se. sylvatica* was found also in Luangnamtha. Further in-depth classification and studies of *Se. perturbans* is needed.

*Sergentomyia phasukae* was first recorded in Laos from Vientiane province [22]. This species was also found in karstic areas of Luangnamtha and Luangphabang. *Se. phasukae* is closely related to *Se. quatei*, only separated from each other by the length of the common spermathecal duct [24,51]. This could lead to misidentification of these two species depending on the slide mounting condition.

We record *Sergentomyia tambori* for the first time in Laos. This species was originally described from Malaysia [24]. However, this species has never been recorded in other countries in Southeast Asia. Specimens found in this study agree with the original description, especially cibarial teeth and spermathecae (Fig 3H, 3I).

**Leishmania-Trypanosoma DNA detection.** This is the first report of *Trypanosoma* DNA being detected in sandflies in Laos. To our knowledge, it is the first time the genera *Chinius* and *Idiophlebotomus* are recorded as carrying parasite DNA. Moreover, the prevalence is high regarding the genus *Idiophlebotomus* (2/25 = 8%) and very high regarding the genus *Chinius* (22/53 = 41.51%). *Trypanosoma* DNA has also been detected in sandflies from Southern Thailand [52]. Though a specific *Trypanosoma* spp. PCR was not used in this study, PCR was able to detect the ITS1 + 5.8S region of *Trypanosoma* spp. by cross amplification. Phylogenetic tree analysis based on these ITS1 + 5.8S sequences showed that *Trypanosoma* DNA from *Ch. eunicegalatiae* and *Id. longiforceps* clustered in two distinct clades with *Trypanosoma* spp. previously identified in sandflies from southern Thailand [41]. These sandfly-derived sister groups were clearly distinct from other known *Trypanosoma cruzi*/*Trypanosoma rangeli*, *Trypanosoma lewisi*, *Trypanosoma theileri*, and *Trypanosoma brucei* clades.

However, because of the high divergence observed between the sequences of the present study and those available in public databases, we cannot draw general conclusions about the taxonomic status of these populations detected by ITS1 + 5.8S sequencing, nor comment on the host specificity and diversity or distribution of the genus *Trypanosoma* in Laos. This uncertainty reflects a broader challenge in *Trypanosoma* systematics: as Stevens and Gibson have emphasized, phylogenetic relationships within the genus are highly complex and difficult to resolve, not only due to the great genetic diversity of the parasites, but also because of issues related to gene choice, alignment ambiguities, and the limitations of single-marker approaches. They stressed the importance of combining multiple genetic markers and phylogenetic methods to obtain more reliable evolutionary inferences [53]. Our study emphasizes a strong specificity between parasites and vectors.

A recent study in Thailand using SSU rRNA [54] linked sandfly-derived *Trypanosoma* to both amphibian-associated and the *Trypanosoma cruzi* clades. Our ITS1 + 5.8S dataset may have missed such associations, since no sequences from amphibian-associated *Trypanosoma* clades (e.g., Anura04/Frog1) were available for comparison. Further studies should therefore incorporate additional genes and *Trypanosoma*-specific PCR assays, along with vertebrate host sampling, to clarify the diversity, host range, and ecological roles of *Trypanosoma* spp. in Laos. A limitation of this study is that the selected sandfly collection locations might not cover areas where *Leishmania* sp. may be circulating. As most areas were not near rural villages but in karstic caves, the captured sandflies may not have been exposed to *Leishmania.* The limited locations and time of year for collection may also have impacted *Leishmania* detection. Further studies should be conducted in wider geographical locations with a higher number of traps per area to collect a greater number of sandflies for a more comprehensive investigation. A previous study conducted in a human leishmaniasis-free area in Thailand also reported negative results for *Leishmania* infection among the collected sandflies. However, in a leishmaniasis-positive area, *Leishmania* DNA was detected among sandflies collected from the house/peri houses of patients [52]. Together with serological surveys in humans and/or animals from different areas of Laos, sandfly studies of greater depth will provide more information on the epidemiological aspect of leishmaniasis in Laos and improve public health mitigation strategies.

## Conclusion

Sandflies in Laos exhibit significant genetic diversity and environmental distribution across seven provinces, with species composition varying by habitat. While no *Leishmania* DNA was detected, the PCR assay did detect *Trypanosoma* DNA, suggesting the first identification in Laos, which will need verification. Studies should be conducted in wider geographic locations with a greater number of traps per area to collect additional sandflies for a more comprehensive investigation.

## Supporting information

**S1 Text. Technical analysis of the *Trypanosoma* phylogenetic tree.**
(DOC)

**S1 Table. Primers used for the detection of Leishmania sp., sandfly cty-b, and blood meal analysis of vertebrate *cyt-b*.**
(XLSX)

**S2 Table. Details and source of sequences used in this study.**
(XLSX)

**S3 Table. The pairwise distance of the cytochrome b gene within species and between species found in this study.**
(XLSX)

**S4 Table. Sandfly species and number used for *Leishmania* spp. screening.** BK: Bokeo, LNT: Luangnamtha, LPB: Luangphabang, VTP: Vientiane province, VTC: Vientiane Capital, and XK: Xiengkhouang.
(XLSX)

**S1 Fig. A full maximum likelihood phylogenetic tree constructed from 222 *cyt-b* sequences of 397 nucleotides.** The tree was constructed using IQ-TREE with model auto selected based on Bayesian Information Criterion (BIC). The numbers on the branches represent the bootstrap values (%) derived from 1000 replicates (options: -m TEST -bb 1000 -alrt 1000 -abayes). Reference sequences of each species generated in previous studies and available in GenBank were selected and included in this analysis. Green color indicates species found in this study. New species that are closely related to *Se. maiae* are labelled in red color.
(TIF)

## Acknowledgments

We thank the local officials at the provincial and district levels and all the villagers who facilitated our research. Institut Pasteur du Laos (IP-Laos) staff provided expert assistance in making arthropod collections.

The views expressed in this article reflect the results of research conducted by the author and do not necessarily reflect the official policy or position of the Department of the Navy, Department of Defense, nor the United States Government. Jodie M. Fiorenzano (LCDR, MSC, USN), Noel Cote (LCDR, MSC, UNS), Irina V. Etobayeva (LCDR, MSC, UNS), and Andrew G. Letizia (CAPT, MC, USN) are military service members of the United States government. This work was prepared as part of their official duties. Title 17 U.S.C. 105 provides that copyright protection under this title is not available for any work of the United States Government.' Title 17 U.S.C. 101 defines a U.S. Government work as work prepared by a military service member or employee of the U.S. Government as part of that person's official duties. This work was prepared as part of their official duties while working at the United States Naval Medical Research Unit INDO PACIFIC

## Author contributions

**Conceptualization:** Khamsing Vongphayloth, Tamalee Roberts, Jodi M. Fiorenzano, Irina V. Etobayeva, Andrew G. Letizia, Philippe Buchy, Paul T. Brey, Jérôme Depaquit.

**Funding acquisition:** Khamsing Vongphayloth, Tamalee Roberts, Jodi M. Fiorenzano, Noel Cote, Irina V. Etobayeva, Matthew T. Robinson, Andrew G. Letizia, Philippe Buchy, Paul T. Brey.

**Investigation:** Khamsing Vongphayloth, Tamalee Roberts, Aphaphone Adsamouth, Khaithong Lakeomany, Veaky Vungkyly, Phonesavanh Luangamath, Somsanith Chonephetsarath, Fano José Randrianambinintsoa, Jérôme Depaquit.

**Project administration:** Khamsing Vongphayloth, Jodi M. Fiorenzano, Noel Cote, Irina V. Etobayeva.

**Supervision:** Andrew G. Letizia, Philippe Buchy, Paul T. Brey, Jérôme Depaquit.

**Visualization:** Khamsing Vongphayloth, Jérôme Depaquit.

**Writing – original draft:** Khamsing Vongphayloth.

**Writing – review & editing:** Tamalee Roberts, Jodi M. Fiorenzano, Irina V. Etobayeva, Matthew T. Robinson, Fano José Randrianambinintsoa, Andrew G. Letizia, Philippe Buchy, Paul T. Brey, Jérôme Depaquit.

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
