## [Decision Letter · Decision Letter 0]

4 Apr 2025

Analysis of Phlebotomine sandflies in Laos from 2014-2024: inventory, description of a new species, screening for Leishmania and detection of Trypanosoma

Dear Dr. VONGPHAYLOTH,

Thank you for submitting your manuscript to PLOS Neglected Tropical Diseases. After careful consideration, we feel that it has merit but does not fully meet PLOS Neglected Tropical Diseases's publication criteria as it currently stands. Therefore, we invite you to submit a revised version of the manuscript that addresses the points raised during the review process.

Please submit your revised manuscript within 60 days Jun 03 2025 11:59PM. If you will need more time than this to complete your revisions, please reply to this message or contact the journal office at plosntds@plos.org. Please include the following items when submitting your revised manuscript:

We look forward to receiving your revised manuscript.

Kind regards,

James Lee Crainey, Ph.D.

Academic Editor

Paul Mireji

Section Editor

Shaden Kamhawi

co-Editor-in-Chief

Paul Brindley

co-Editor-in-Chief

**Journal Requirements:**

At this stage, the following Authors/Authors require contributions: Khamsing VONGPHAYLOTH, Tamalee Roberts, Jodi M. Fiorenzano, Noel Cote, Irina V. Etobayeva, Aphaphone Adsamouth, Matthew T. Robinson, Khaithong Lakeomany, Veaky Vungkyly, Phonesavanh Luangamath, Somsanith Chonephetsarath, Fano José Randrianambinintsoa, Andrew G. Letizia, Philippe Buchy, Paul T. Brey, and Jérôme Depaquit. Please ensure that the full contributions of each author are acknowledged in the "Add/Edit/Remove Authors" section of our submission form.

- ® on page: 9.

Potential Copyright Issues:

- Figure 1. Please provide a direct link to the base layer of the map (i.e., the country or region border shape) and ensure this is also included in the figure legend; and provide a link to the terms of use / license information for the base layer image or shapefile. We cannot publish proprietary or copyrighted maps (e.g. Google Maps, Mapquest) and the terms of use for your map base layer must be compatible with our CC BY 4.0 license.

6) Please ensure that the funders and grant numbers match between the Financial Disclosure field and the Funding Information tab in your submission form. Note that the funders must be provided in the same order in both places as well.

**Reviewers' Comments:**

Reviewer's Responses to Questions

**Key Review Criteria Required for Acceptance?**

**Methods:**

-Are the objectives of the study clearly articulated with a clear testable hypothesis stated?

-Is the study design appropriate to address the stated objectives?

-Is the population clearly described and appropriate for the hypothesis being tested?

-Is the sample size sufficient to ensure adequate power to address the hypothesis being tested?

-Were correct statistical analysis used to support conclusions?

-Are there concerns about ethical or regulatory requirements being met?

Reviewer #1: The manuscript "Analysis of Phlebotomine sandflies in Laos from 2014-2024: inventory, description of a new species, screening for Leishmania and detection of Trypanosoma" is very interesting as it provides data from a place where little is known about both on sandflies and sandfly-borne pathogens. The study aimed to report species diversity data and Leishmania detection among sandflies collected from seven provinces of Laos. The objectives of the study are clearly articulated and properly designed, based on their assumption that the bordering country to Laos, Thailand has reported autochthonous cases of leishmaniases. 3,857 sandflies from 25 species belonging to five genera were collected for nearly a decade (2014-2024) and identified morphologically and molecularly. They performed PCR-technique targetting the ITS1 region to screen for Leishmania in the sandfly samples and by direct nucleotide sequencing. The methods are properly described and the sample size of sandflies collected is suitable to address their hypothesis. The study does not need to perform statistical analysis. However, proper software were used for phylogenetic analysis.

Reviewer #2: Please see my General Comments.

Reviewer #3: The methods are okay, but the phylogenetics was poorly done. It's questionable whether the primer for blood-meal analysis were appropriate. The authors should be performing morphometric analysis on their sandflies.

Reviewer #4: The manuscript presents adequately described the objetives. Being a an ecological descriptive study sample size is not required.

**Results:**

-Does the analysis presented match the analysis plan?

-Are the results clearly and completely presented?

-Are the figures (Tables, Images) of sufficient quality for clarity?

Reviewer #1: The results are properly presented and match the analysis plan. The figures are of good quality.

Reviewer #2: Please see my General Comments.

Reviewer #3: Potentially interesting: the authors obtained 3 positive pools from 221 pools comprising a few thousand flies. It is a huge shame the blood meal analysis failed and I suggest the authors try again and do a full literature review of

The Tryp sp. tree was incomprehensible. There needs to be landmark trips to see what the species might be close to. I would suggest creating a rich species dense tree around all sequences associated with T. Rangel, which will include T. cruzi. Just about all these genomes have been sequenced so extracting the ITS should be easy. The issue is I can see a tree, full of Trypanosome sp. and I have no idea what it's saying. I recognise T. avium and of course T. rangel but thats about it. Thats not very informative of the species rich diversity of Tryps. Ultimately 3 tryps form a few thousand sandflies doesn't really jump out at me, but its interesting. If the blood meal analysis identified the specific vertebrates/mammals then that starts looking much more interesting. If the authors read the historic papers by Stevens et al and I would assume the genome trp trees after this (if they ever finished them), that is diversity of tryps you need for a macro-scale tryp tree and will help root the subsequent species specific tree: then a smaller tree correctly rooted from the macro-scale tree focusing around T. rangel and every sequence on Genbank therein. Do a series of Blasts via teh NCBI website - the entire sequences can be automatically downloaded by "cutting" the flanking loci of the hits. This will then need aligning (e.g. muscle5) and then building a tree via IQTREE. Make sure the alignment is correctly manually edited via appropriate software.

I can not comment on sandflies. I can't interpret the tree, albeit the methods are not state of the art, but thats relatively easy to correct. The authors should make their own investigation into this (guidance below) and https://bioinformatics.stackexchange.com may provide support.

Reviewer #4: Result are clearly described and in concordance with the objetives and methods described. All figures and tables are necesary.

**Conclusions:**

-Are the conclusions supported by the data presented?

-Are the limitations of analysis clearly described?

-Do the authors discuss how these data can be helpful to advance our understanding of the topic under study?

-Is public health relevance addressed?

Reviewer #1: Yes to all

Reviewer #2: Please see my General Comments.

Reviewer #3: The conclusions are interesting, but underwhelming.

Basically if the sandfly experts think the authors have an interesting observations on their new species I am happy to look at providing extensive revisions of the manuscript, but at present there are a lot of rough edges here. Again, if there were 24 decisive blood meal identifications the authors would be in a strong position to claim a paper at PLoS NTD

Reviewer #4: Conclussion are adequate to the objetive of the study.

**Editorial and Data Presentation Modifications?**

Reviewer #1: This is an interesting manuscript and provides data from a place where little is known about the diversity of the sandfly population. Although the aim of the study was also to detect the presence of Leishmania parasites based on the assumption that the bordering country, Thailand that has reported cases of leishmaniasis, the study contributes to the literature by showing the absence of Leishmania but the presence of Trypanosoma. I suggest it is interesting to publish the study.

Reviewer #2: Please see my General Comments.

Reviewer #3: The tryp tree labels are incomprehensible and need a graphics designer to help the authors. That tree is not just not publication standard, its review standard, again I cannot read some of the species labels they are so blurred.

Reviewer #4: A minor modification is suggested for lines 49-53 for be more clear.

**Summary and General Comments:**

Reviewer #1: This is an interesting manuscript and provides data from a place where little is known about the diversity of the sandfly population. Although the aim of the study was also to detect the presence of Leishmania parasites based on the assumption that the bordering country, Thailand that has reported cases of leishmaniasis, the study contributes to the literature by showing the absence of Leishmania in the region studied but the presence of Trypanosoma. Unfortunately, the authors did not go further to identify the Trypanosoma species.

Reviewer #2: Dear Editor,

In this study, Vongphayloth and colleagues investigated the diversity of phlebotomine sand flies in Laos. The manuscript presents a large-scale field study with a remarkable taxonomic effort, identifying 3,857 specimens belonging to 25 different phlebotomine sand fly species. In addition to classical morphological taxonomy, the authors employed a PCR-based method to amplify a fragment of the cytochrome b gene for molecular species confirmation. Through both approaches, they observed morphological and molecular differences suggesting the presence of a previously undescribed phlebotomine sand fly species. Consequently, they describe Sergentomyia sutherlandi n. sp., integrating both morphological and molecular traits. Finally, the authors attempted to detect Leishmania sp. DNA in the collected sand flies, which was unsuccessful, but they did report the presence of Trypanosoma sp. DNA in a few specimens.

I find this study highly valuable to the scientific community. Beyond describing a new species of phlebotomine sand fly, this study may provide genetic data for several species that are challenging to identify morphologically and could contribute to discussions on their taxonomic status.

While this version of the manuscript is well-written and presents solid research, I have some comments that the authors may find helpful in further improving and clarifying certain points.

Introduction:

This section is consistent and well-structured.

Methods:

Sand fly cytochrome b gene database:

• Please cite the sequencing instrument used.

• Ensure that the collection site location is included in the GenBank features (i.e., geo_loc_name). This is an important detail often overlooked when submitting sequences to GenBank.

Blood meal analysis:

• Did the authors include a PCR positive control for this analysis? Since no detection was reported (Results, line 353), I wonder whether this outcome was due to a late-stage blood meal that had already been digested (most likely) or a technical issue.

Molecular detection of Leishmania and Trypanosoma species in sand flies:

• Was a PCR positive control included?

Phylogenetic analysis of sand fly species and detected parasites:

• Please double-check the type of phylogenetic inference used by IQ-TREE. To the best of my knowledge, this tool employs maximum-likelihood inference rather than Bayesian inference.

Results:

Construction of cyt-b database and phylogenetic analysis of sand flies:

• Lines 220–225: This detailed information seems excessive, as the number of sequences per species is already provided in Table S2.

• Figure 2 (phylogenetic tree): Details on the criteria used to select sequences from GenBank would improve data interpretation for readers.

o I recommend including a full-view tree (i.e., without collapsing nodes per species/groups) as a supplementary file. Please ensure that accession codes for all sequences are included.

o Lines 240–242: What is the authors’ interpretation of the low pairwise distance between Se. rudnicki and Se. brevicaulis? Since these two species are collapsed into the Se. brevicaulis group clade, it is difficult to interpret this result. A full-view tree or a revised version showing the clade with nominal species might clarify this.

o Phlebotomus argentipes is colored in green (indicating a sequence generated in this study), but Lines 412–413 state that Ph. argentipes was not recorded, only specimens identified as Ph. argentipes-like. Please clarify this inconsistency.

Description of new sand fly taxa:

I do not feel comfortable commenting on or judging the taxonomic traits used for the new species description.

However, with the advent of molecular tools, I would recommend generating additional reference nucleotide sequences for key genetic markers in new species descriptions. While not mandatory, this is a good practice that can facilitate future discussions and reassessments of species taxonomy. In addition to cyt-b, other mitochondrial markers such as cox1, ND4, ITS1, 16S rRNA, and 12S rRNA might be considered for this and future descriptions.

Leishmania detection:

• Please clarify the criteria for including GenBank sequences.

• A best-hit table with BLAST results could be beneficial for Trypanosoma sp. molecular identification.

• Figure 6:

o Ensure that the legend follows the same format as the phlebotomine tree.

o Please double-check the description for the sequence OL332783. In the original publication, this sequence is linked to Sergentomyia sp., whereas in your tree, it appears as Id. asperulus. In the main text, you refer to Sergentomyia sp. Please correct this accordingly.

Blood meal detection:

• Line 354: “A total of 29 semi-gravid females”. However, in Methods (Line 162), it states: “Engorged sand fly samples were used to determine the blood meal origin.” Do the authors consider these terms synonymous? Please revise for consistency.

• A positive control is required to confirm that the PCR assay worked as expected.

Discussion:

• Lines 363–371: This information is already reported in the Results section.

• Lines 387–388: What environmental conditions in karstic areas may promote optimal sand fly incidence?

• Lines 445 and 450: Could the authors comment on the species complexes Sergentomyia anodontis s.l. and Sergentomyia bailyi s.l.? What other species could be hidden or misidentified under these nominal species? Could additional genetic markers help distinguish them?

• Line 460: Is Se. gemmea-like represented by the sequence “Se. sp 1” in the phylogenetic tree? If so, please revise accordingly.

• Line 469: If any sequence of Se. quatei is available, please include it.

Reviewer #3: I am underwhelmed by the results, but they are interesting. The issue is whether 0.1% positivity is epidemiologically meaningful. Without blood-meal analysis it's impossible for someone else to followup the study. Thus normally the mammals, birds would be trapped and sampled for the tryps - in a followup study. The situation is if I could be certain that sandflies were transmitting tryps thats huge news. However, I cannot be certain it's not incidental feeding, i.e. the tryps will not propagate in the sandfly nor be transmitted. That's the central issue here, there's not enough positives to exclude incidental feeding but equally 0.1% is an epidemiologically acceptable prevalence in a flying vector. Then if we ask the question, well the flies may not be heavily infected but the mammals or birds will have high tryp prevalence; reservoir hosts or tryps can have extreme prevalence of infection. The issue here is we don't know what the sandflies are feeding on, so there's no chance of a follow-up study. Alternatively if the authors had obtained tryp isolates - that would be interesting because they can exist perpetually if carefully cultured and stored.

In summary, strictly from my point of view - excluding the importance of a new species of sandfly - there's not enough here to say this is a PLoS NTD paper. So I will leave this to the entomologists to guidance on the suitability of paper for the journal.

Reviewer #4: The manuscript by Vongphayloth et al. presents the results of an interesting study describing the sand fly fauna in Laos. The study is important due to the scarcity of information on this issue in the region. Considering the role of some sand fly species in transmitting protozoan parasites such as leishmania, the results contribute new information for the surveillance of this parasitic disease. The findings also presented the description of a new species to science, increasing the knowledge of the diversity of this insect group.

lines 49-53: Studies should be conducted in a wider geographical location with a higher number of traps per area in order to collect a greater number of sand flies to better define this vector and its pathogens. I recommend reviewing this sentence, because collecting a large number of sand flies is not sufficient for vector definition. I suggest modifications to: “Studies should be conducted, increasing the sampling effort to analyze population dynamics and other vector capacity parameters to determine the vector species and pathogen circulation.”

The introduction is clear and provides the necessary information to understand the problem. The methods were broadly described and are understandable. The results show interesting novelty with the description of the diversity of sand flies and trypanosomes in the region, and the description of a new species to science. The discussion is adequately based on the results and previous literature reports. The conclusions are adequate with the modification suggested below.

Best regards,

PLOS authors have the option to publish the peer review history of their article (what does this mean? ). If published, this will include your full peer review and any attached files.

**Do you want your identity to be public for this peer review?** For information about this choice, including consent withdrawal, please see our Privacy Policy .

Reviewer #1: **Yes: ** Rajendranath Ramasawmy

Reviewer #2: **Yes: ** Lucas Sousa-Paula

Reviewer #3: No

Reviewer #4: No

**Figure resubmission:**

**Reproducibility:**



---

## [Decision Letter · Decision Letter 1]

28 Jun 2025

Analysis of Phlebotomine sandflies in Laos from 2014-2024: inventory, description of a new species, screening for Leishmania and detection of Trypanosoma

Dear Dr. VONGPHAYLOTH,

Thank you for submitting your manuscript to PLOS Neglected Tropical Diseases. After careful consideration, we feel that it has merit but does not fully meet PLOS Neglected Tropical Diseases's publication criteria as it currently stands. Therefore, we invite you to submit a revised version of the manuscript that addresses the points raised during the review process.

Please submit your revised manuscript within 30 days Jul 28 2025 11:59PM. If you will need more time than this to complete your revisions, please reply to this message or contact the journal office at plosntds@plos.org. Please include the following items when submitting your revised manuscript:

* A rebuttal letter that responds to each point raised by the editor and reviewer(s). You should upload this letter as a separate file labeled 'Response to Reviewers '. This file does not need to include responses to any formatting updates and technical items listed in the 'Journal Requirements' section below.

* A marked-up copy of your manuscript that highlights changes made to the original version. You should upload this as a separate file labeled 'Revised Manuscript with Track Changes '.

* An unmarked version of your revised paper without tracked changes. You should upload this as a separate file labeled 'Manuscript '.

We look forward to receiving your revised manuscript.

Kind regards,

James Lee Crainey, Ph.D.

Academic Editor

Paul Mireji

Section Editor

Shaden Kamhawi

co-Editor-in-Chief

Paul Brindley

co-Editor-in-Chief

**Journal Requirements:**

1) We do not publish any copyright or trademark symbols that usually accompany proprietary names, eg ©,  ®, or TM  (e.g. next to drug or reagent names). Therefore please remove all instances of trademark/copyright symbols throughout the text, including:

- TM on page: 11.

 2) Please ensure that the funders and grant numbers match between the Financial Disclosure field and the Funding Information tab in your submission form. Note that the funders must be provided in the same order in both places as well. Currently, the Financial Disclosure states there was no funding received.

3) Thank you for stating 'XXXX' Please note that, though access restrictions are acceptable now, your entire minimal dataset will need to be made freely accessible if your manuscript is accepted for publication. This policy applies to all data except where public deposition would breach compliance with the protocol approved by your research ethics board. If you are unable to adhere to our open data policy, please kindly revise your statement to explain your reasoning and we will seek the editor's input on an exemption.

4) Sequences of Sandfly Cyt-b obtained from this study were deposited in GenBank under accession numbers PV054613– PV054752). Sequences of Trypanosoma spp. detected from this study were deposited in the GenBank database (PV034524 - PV034547). Type specimens. The holotype female (voucher SF22-EX651) of Sergentomyia sutherlandi n. sp. and one female paratype (voucher SF22-EX695) have been deposited at the Terrestrial Arthropods Collection of the Muséum national d’Histoire naturelle (MNHN, Paris) under the inventory numbers MNHN-ED-11547 and MNHN-ED-11548.

**Reviewers' comments:**

Reviewer's Responses to Questions

**Key Review Criteria Required for Acceptance?**

**Methods**

-Are the objectives of the study clearly articulated with a clear testable hypothesis stated?

-Is the study design appropriate to address the stated objectives?

-Is the population clearly described and appropriate for the hypothesis being tested?

-Is the sample size sufficient to ensure adequate power to address the hypothesis being tested?

-Were correct statistical analysis used to support conclusions?

-Are there concerns about ethical or regulatory requirements being met?

Reviewer #2: The methods were performed to answer the objectives of the study.

Reviewer #3: The authors have done a reasonable job of their tryp trees; specifically they were asked to get a new representative sample distribution so a "tryp specialist" can interpret what type of tryps the authors have detected. However, the phylogenetic methods need a far more detailed explanation. I would recommend the commands they used are presented in the appendix. Furthermore, the alignment has not been manually inspected. Manual curation and certification is essential and was not described. Therefore I would strongly suggest this is performed and where necessary the trees re-calculated. This applies to both sandfly and tryp trees. The congruence between MEGA and IQTREE needs reporting and BOTH sets of bootstraps should be represented on the trees. It's not clear which tree, nor which bootstraps the authors have presented.

**Results**

-Does the analysis presented match the analysis plan?

-Are the results clearly and completely presented?

-Are the figures (Tables, Images) of sufficient quality for clarity?

Reviewer #2: The authors have followed the reviewers' recommendations. The current version if the manuscript has been improved.

Reviewer #3: The key criticism is that the trees have substantially below publication, moreover the tryp tree has key information which I am unable to read. Specifically I want to know which tryps the authors sequences form sister groups with, i.e. which insects were the database sequences recovered. The key question is whether these are incidental infections or an epidemiology (between sandflies and tryps) which has yet to be formalised. The sister group insect vector of the authors sequences will give an indication. There is no option but for the authors to perform a new revision and return this for inspection, because the tree is not intelligible.

Secondly, the authors have poorly described the location within the tree of their tryp sequences. T. sp. isn't a tryp. Again we are looking for overall positioning, i.e. rodent versus T. rangeli/ T. cruzi clades and other information concerning vector/vertebrate isolation data. This needs to be professionally described within the results section and it significantly below the required standard. Again, what we're assessing is whether their is evidence supporting or rejecting incidental infection or new epidemiological cycle.

PLoS NTD is an internationally recognised journal of high standing and half-baked descriptors are not acceptable in this journal.

**Conclusions**

-Are the conclusions supported by the data presented?

-Are the limitations of analysis clearly described?

-Do the authors discuss how these data can be helpful to advance our understanding of the topic under study?

-Is public health relevance addressed?

Reviewer #2: Yes.

Reviewer #3: The discussion has been handled nicely in identifying the weaknesses/limitations of the study. The authors have omitted citations by Stevens/Gibson (tryps) and Ready (sandflies). This should be addressed; was their tree congruent with the phylogenies of Stevens/Hamilton/Gibson? Furthermore although Ready focused on sandflies in north African counties (possibly the Middle East) this investigator was an early exponent of cyt B taxonomic identification and population genetics. These issues require addressing in the discussion. The citation for Dvorak was appropriate.

**Editorial and Data Presentation Modifications?**

Reviewer #2: Line 64: Since there is no information about this Trypanosoma spp. pathogenicity, I recommend deleting “pathogens.”

Line 77: Please italicize Leishmania, and include “parasites” (i.e., Leishmania parasites).

Lines 138–148 (Fig 1): Please ensure to mention that the red stars refer to the collection sites.

Line 180: “spp.” should not be italicized.

Line 198: Please cite the origin of the human blood source. Was the DNA extraction method used the same as for sand flies? Please note that mentioning “human blood” may raise ethical concerns. If the authors used a DNA sample from a previous study, I suggest mentioning that instead, along with the origin of the material.

Lines 205–206: Please cite the origin of L. major and L. tropica DNA (e.g., cultured strain + strain number).

Line 226: Please insert a comma after "508 males."

Tables 2 and 3: Please also include the abbreviations of genera in the notes.

Line 511: “Complex species group” should be corrected to “species group” or “species complex.” Both terms are used to refer to informal assemblages of taxa, particularly among phlebotomine sand flies (PMID: 33480064; DOI: 10.1186/s13071-025-06748-5).

Lines 514-517: “However, after more recent studies, species reported in this region may include Se. hivernus, Se. dvoraki, Se. khawi, and Se. gemmea [20]. However, the records of Se. gemmea in Laos are pending revision, as the pairwise distance of the cyt-b gene between specimens from Laos and Thailand was only 0.06 (6%) [14].”

This is a bit confusing. Do the authors consider the pairwise distance of 6% too low? As written, it sounds like the molecular distance of 6% might indicate different species (Laos vs. Thailand), but the morphological difference observed did not meet the criteria used by the authors for species description.

Line 519: Please consider replacing “to have” with “to exhibit” and “needed” with “necessary.”

Line 519: For better reading flow, I suggest splitting this sentence: “…for a new species. Instead, we will await…”

Line 532: Please use “Southeast Asia” consistently throughout the text for uniformity.

Reviewer #3: What has been presented by the authors is reasonably technically sound but significantly below acceptable standard in presentation and interpretation.

**Summary and General Comments**

Reviewer #2: The authors have followed the reviewers recommendations and resubmitted an improved version of the manuscript.

Reviewer #3: Overall, the authors have got onto the next rung of the ladder, but this submission at present is not currently of publication standard, but sufficiently technically robust to continue toward the next revision.

The final point is data availability the sequences MUST be submitted to Genbank or EBI for the next revision of the manuscript. The authors can delay release of the sequences on acceptance of the ms, but NCBI numbers (or EBI) for accession must be presented in the next draft of the ms both for tryps and sandlflies. Under the terms of PLoS NTD publication criteria this is mandatory. I have not personal interest in the data, but other investigators may want to examine the data in the future.

PLOS authors have the option to publish the peer review history of their article (what does this mean? ). If published, this will include your full peer review and any attached files.

**Do you want your identity to be public for this peer review?** For information about this choice, including consent withdrawal, please see our Privacy Policy .

Reviewer #2: No

Reviewer #3: No

**Figure resubmission:**
---

## [Decision Letter · Decision Letter 2]

13 Oct 2025

Dear Dr. VONGPHAYLOTH,

We are pleased to inform you that your manuscript 'Analysis of Phlebotomine sandflies in Laos from 2014-2024: inventory, description of a new species, screening for Leishmania and detection of Trypanosoma' has been provisionally accepted for publication in PLOS Neglected Tropical Diseases.

Best regards,

James Lee Crainey, Ph.D.

Academic Editor

Paul Mireji

Section Editor

Shaden Kamhawi

co-Editor-in-Chief

Paul Brindley

co-Editor-in-Chief

Reviewer's Responses to Questions

**Key Review Criteria Required for Acceptance?**

**Methods**

-Are the objectives of the study clearly articulated with a clear testable hypothesis stated?

-Is the study design appropriate to address the stated objectives?

-Is the population clearly described and appropriate for the hypothesis being tested?

-Is the sample size sufficient to ensure adequate power to address the hypothesis being tested?

-Were correct statistical analysis used to support conclusions?

-Are there concerns about ethical or regulatory requirements being met?

Reviewer #3: There's significant improvement in the bioinformatics in the paper and the authors have reached an acceptable standard.

**Results**

-Does the analysis presented match the analysis plan?

-Are the results clearly and completely presented?

-Are the figures (Tables, Images) of sufficient quality for clarity?

Reviewer #3: The adequately described their phylogenies within the standards required

**Conclusions**

-Are the conclusions supported by the data presented?

-Are the limitations of analysis clearly described?

-Do the authors discuss how these data can be helpful to advance our understanding of the topic under study?

-Is public health relevance addressed?

Reviewer #3: It's an interesting study. The next step is to obtain isolates

**Editorial and Data Presentation Modifications?**

Reviewer #3: NA

**Summary and General Comments**

Reviewer #3: The bioinformatics is not perfect but the authors have made a significant effort to reach an acceptable standard. It is an interesting result and warrants follow-up studies, which I hope to see published here. It's potentially a revision of Trypanosoma sp. vector transmission, but it needs further investigation

PLOS authors have the option to publish the peer review history of their article (what does this mean? ). If published, this will include your full peer review and any attached files.

**Do you want your identity to be public for this peer review?** For information about this choice, including consent withdrawal, please see our Privacy Policy .

Reviewer #3: **Yes: ** Michael W. Gaunt

---

## [Editor Report · Acceptance letter]

Dear Dr. VONGPHAYLOTH,

We are delighted to inform you that your manuscript, "Analysis of Phlebotomine sandflies in Laos from 2014-2024: inventory, description of a new species, screening for Leishmania and detection of Trypanosoma," has been formally accepted for publication in PLOS Neglected Tropical Diseases.

Best regards,

Shaden Kamhawi

co-Editor-in-Chief

Paul Brindley

co-Editor-in-Chief
